# Clinical utility of polygenic scores for cardiometabolic disease in Arabs

Injeong Shim [1,2,3,4,11], Hiroyuki Kuwahara[5,6,11], NingNing Chen [5,6,11], Mais O. Hashem[7], Lama AlAbdi [7,8], Mohamed Abouelhoda[9], Hong-Hee Won[4], Pradeep Natarajan [1,2,3], Patrick T. Ellinor [1,3], Amit V. Khera[10], Xin Gao [5,6,12 ✉], Fowzan S. Alkuraya [7,12 ✉] & Akl C. Fahed [1,2,3,12 ✉]

Arabs account for 5% of the world population and have a high burden of cardiometabolic disease, yet clinical utility of polygenic risk prediction in Arabs remains understudied. Among 5399 Arab patients, we optimize polygenic scores for 10 cardiometabolic traits, achieving a performance that is better than published scores and on par with performance in European-ancestry individuals. Odds ratio per standard deviation (OR per SD) for a type 2 diabetes score was 1.83 (95% CI 1.74–1.92), and each SD of body mass index (BMI) score was associated with 1.18 kg/m² difference in BMI. Polygenic scores associated with disease independent of conventional risk factors, and also associated with disease severity—OR per SD for coronary artery disease (CAD) was 1.78 (95% CI 1.66–1.90) for three-vessel CAD and 1.41 (95% CI 1.29–1.53) for one-vessel CAD. We propose a pragmatic framework leveraging public data as one way to advance equitable clinical implementation of polygenic scores in non-European populations.

Polygenic scores can identify individuals at risk of disease, but their use in clinical practice is limited by the lack of widely accepted standards and reduced cross-ethnic transferability[1–5]. Despite many statistical methods and published scores, there is no clear framework to guide a new population interested in implementing polygenic scores using this publicly available data. Cross-ethnic transferability of scores—mostly derived from individuals of European ancestry—to other populations who are less represented in genome-wide association studies (GWAS) also suffers from reduction in performance, but new computational methods are improving on this limitation for Asian, African, and other ancestries[1–4,6,7].

While the case for clinical utility of polygenic scores—mostly for cardiometabolic disease and some cancers—has been made in European-ancestry populations in the U.S. and Europe[8–11], it is equally important to understand whether the prospect of clinical utility is also relevant to other populations, where genetic ancestry, environmental factors, and disease epidemiology might differ[12]. A recent statement from the American Society of Human Genetics highlighted the

[1] Cardiovascular Research Center, Department of Medicine, Massachusetts General Hospital, Harvard Medical School, Boston, MA, USA. [2] Center for Genomic Medicine, Department of Medicine, Massachusetts General Hospital, Harvard Medical School, Boston, MA, USA. [3] Cardiovascular Disease Initiative, Broad Institute of MIT and Harvard, Cambridge, MA, USA. [4]Department of Digital Health, Samsung Advanced Institute for Health Sciences & Technology, Sungkyunkwan University, Samsung Medical Center, Seoul, South Korea. [5] Computational Biosciences Research Center (CBRC), King Abdullah University of Science and Technology (KAUST), Thuwal, Saudi Arabia. [6]Computer Science Program, Computer, Electrical and Mathematical Sciences and Engineering Division, King Abdullah University of Science and Technology (KAUST), Thuwal, Saudi Arabia. [7] Department of Translational Genomics, Center for Genomic Medicine, King Faisal Specialist Hospital and Research Center, Riyadh, Saudi Arabia. [8] Department of Zoology, College of Science, King Saud University, Riyadh, Saudi Arabia. [9]Department of Computation Sciences, Center for Genomic Medicine, King Faisal Specialist Hospital and Research Center, Riyadh, Saudi Arabia. [10]Verve Therapeutics, Cambridge, MA, USA. [11]These authors contributed equally: Injeong Shim, Hiroyuki Kuwahara, NingNing Chen. [12]These authors jointly supervised this work: Xin Gao, Fowzan S. Alkuraya, Akl C. Fahed. ✉e-mail: xin.gao@kaust.edu.sa; FAlKuraya@kfshrc.edu.sa; afahed@mgh.harvard.edu

problem of reduced portability as a key priority area in human genetics research[12].

Arabs represent about 5% of the world population and are massively under-represented in genomic studies worldwide, yet minimal efforts have been made to date to understand clinical utility of polygenic scores in this group[13,14]. Efforts to understand performance and potential utility of polygenic scores in Arabs are important for two reasons. First, Arabs represent a large and diverse ethnic group inhabiting the Middle East and North Africa, and are present as a diaspora in the United States and Western Europe[13]. Arab countries are also among those with the largest population growth worldwide[15]. Second, there is a large burden of cardiometabolic disease among Arabs with some of the countries in the Arabian peninsula having the highest rates of diabetes and obesity worldwide[16–24]. While conventional risk factors such as poor diet, smoking and sedentary lifestyle are highly prevalent, they incompletely capture excess risk; therefore, it is not clear whether genomic risk may contribute and augment risk identification.

Prior genomic studies for cardiometabolic disease in Arabs are limited due to small (i.e., typically <10,000 participants) genome-wide association studies, no prior efforts to optimize polygenic scores, and most studies being conducted on heterogeneous ethnic Arabs of diverse genetic ancestries[25–30]. The most comprehensive effort from the Qatar Biobank recently reported differences in linkage disequilibrium and effect sizes in a GWAS of 45 traits and showed that European-derived polygenic scores have reduced performance[28]. The study population represented the wider Middle Eastern region with only 37.6% "general" Arabs and 17.3% "Peninsular" Arab, but there was no analysis of score performance in those subpopulations[28]. In this study, we leveraged recent GWAS data and novel multi-ethnic computational methods to optimize polygenic scores for cardiometabolic disease and define their clinical utility in a cohort of indigenous Arabs from Saudi Arabia consisting of 5399 patients and 1017 population reference participants. In using publicly available data, we define a framework for optimizing polygenic scores that could be transposable to other populations.

## Results

### Pragmatic framework for optimizing polygenic scores to a new population

We propose a pragmatic approach to optimize polygenic scores for a new population that leverages available public datasets and recent advances in computational methods (Fig. 1). With the growth of large genomic datasets, there is an important focus on their disproportionate enrichment for individuals of European ancestry and an urgent need for more non-European representation[1,14]. However, less attention is paid to smaller subpopulations beyond the continental ancestries and populations with different environmental factors, where portability of polygenic scores might also be limited. Even in the absence of large genomic datasets from those target subpopulations, we show that a pragmatic framework consisting of four key steps could enable successful optimization for indigenous Arabs.

First, multiple scores are derived using public datasets. The target population genomic data undergoes standard quality control, and imputation for genotyping array data is performed using publicly available imputation panels, prioritizing ancestry-specific (if present) or large and diverse panels. Summary statistics from the largest and most diverse GWAS for the trait of interest are obtained, and multiple scores are derived using five scoring methods—a baseline method (PRSice-2), as well as methods that factor in the genetic architecture (LDpred2, lassosum2, and PRS-CS) and ancestry (PRS-CSx)[6,31–34]. A fixed set of single nucleotide polymorphisms (SNPs) from the target population is used to calculate principal components (PCs) of ancestry which are used to adjust the raw polygenic scores as described previously[7,35].

Second, the target population is split into training and validation sets. The performance of the different scores for a trait is compared in

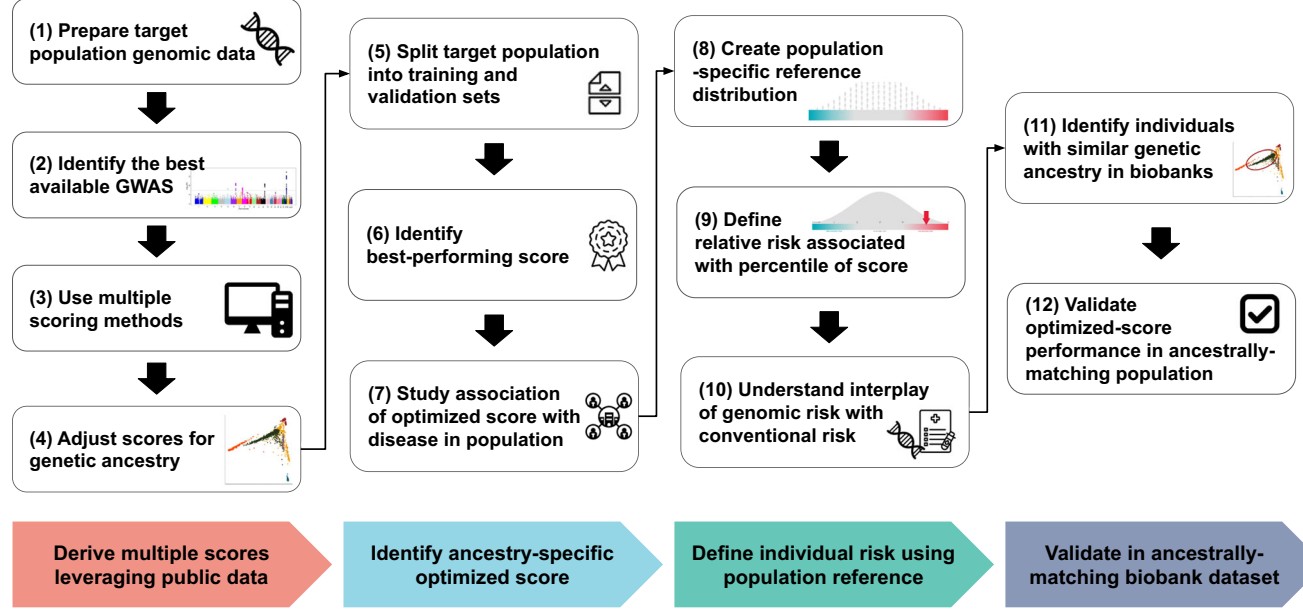

**Fig. 1 | Framework for optimizing polygenic scores for a new population.** A pragmatic framework for optimizing polygenic scores for a target population using publicly accessible datasets and methods consists of four steps. First, genomic data is prepared using standard quality control and imputation to obtain multiple polygenic scores based on available large and diverse GWAS results and various score derivation methods. The raw scores are adjusted for population structure using principal components (PCs) of ancestry. Second, a best-performing score is identified by splitting the dataset into training and validation sets to determine the best model in the training set and to assess the association between ancestry-specific optimized scores and traits in the validation set. Third, individual risk percentile rank is derived from the distribution based on the reference population of the same ancestry in order to identify individual relative risk levels and study the interplay between genetic risk and conventional risk factors. Fourth, to validate the ancestry-specific optimized scores, ancestry-matched samples can be identified in a large biobank dataset using genetic distance.

the training set, and the best-performing score is selected and reported in the validation set where its association with the trait of interest is reported using regression models.

Third, since a polygenic score is best represented as percentile, we propose having a static population reference distribution on which polygenic score percentiles are defined. In the case of Arabs, we use a cohort of 1017 unrelated individuals who self-identify with 28 major clans/tribes in Saudi Arabia and are representative of the general population (Supplementary Fig. 1)[36]. All optimized scores are calculated in this population and used to define percentiles for each score. Subsequently, percentiles of risk are defined based on the population reference distribution, and a relative risk is assigned to each percentile of the score. Downstream analysis enables understanding of the interplay between the risk associated with genomics and conventional risk factors.

Fourth, we propose a method for identifying ancestrally matched individuals to the target sub-population in large biobanks—for example, we could identify participants in the UK Biobank who are ancestrally matched to indigenous Arabs from Saudi Arabia[37]. We demonstrate how such a dataset could be used to extend the optimization of polygenic scores in one country to individuals of a similar ancestry who are living in other parts of the world such as the United States or Europe.

## European-derived polygenic scores have reduced performance in Arabs

Most published polygenic scores are derived from datasets with predominantly individuals of European ancestry. In the Polygenic Score Catalog, 98.0% (1772 of 1808) published scores were trained in datasets that had at least 80% European ancestry distribution[38]. We first asked whether the most commonly used polygenic scores for cardiometabolic disease have reduced performance in Arabs (Fig. 2).

In the cohort of 5399 indigenous Arabs referred for cardiovascular care and as such a high prevalence of cardiometabolic disease (Table 1), we computed the most commonly used polygenic score for coronary artery disease (CAD), type 2 diabetes, cardiomyopathy, and seven continuous traits—low-density lipoprotein (LDL) cholesterol, high-density lipoprotein (HDL) cholesterol, triglycerides (TG), systolic blood pressure (SBP), diastolic blood pressure (DBP), body mass index (BMI), and height (Supplementary Table 1). For each of the 10 traits of interest, we used one-to-one nearest neighbor matching to generate a cohort of individuals of European ancestry from the UK Biobank of equal size as the validation dataset of Arabs, with comparable age, sex, and case-control ratio for categorical matching traits or mean value for continuous matching traits (Methods, Supplementary Table 2).

Comparing the performance of polygenic scores in indigenous Arabs vs. Europeans from the UK Biobank, we observed a substantial

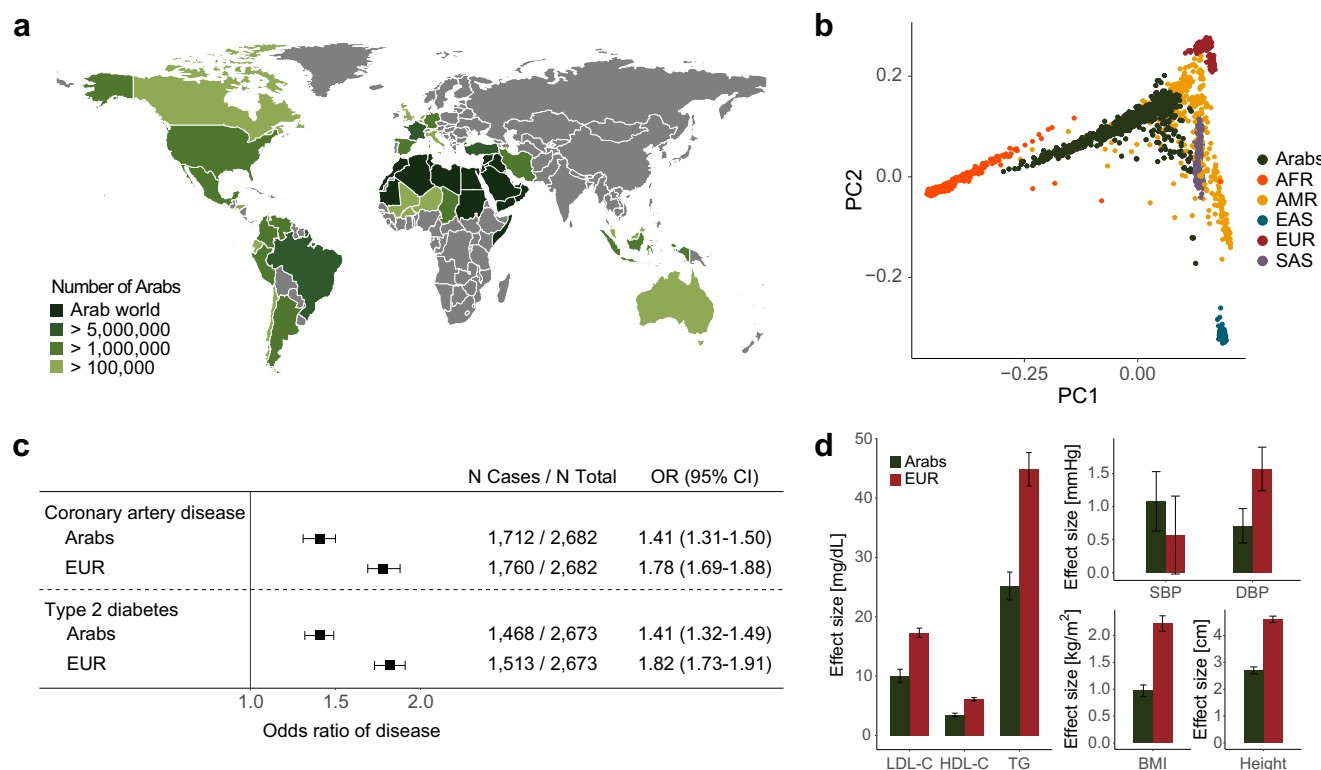

**Fig. 2 | European-derived polygenic scores have reduced performance in Arabs.**
**a** Map showing the distribution of Arab populations around the world (created using the 'rnaturalearth' package in R and the cited data sources)[13, 57, 58]. **b** Principal components of ancestry plot showing indigenous Arabs (in dark green) in this study compared to 1000 Genomes Project populations (AFR: African (in orange); AMR: Admixed American (in yellow); EAS: East Asian (in dark blue); EUR: European (in dark red); SAS: South Asian (in violet)). **c** Performance of polygenic scores for coronary artery disease and type 2 diabetes in Arabs from this study vs. a matched case-control sample of European-ancestry individuals from the UK Biobank (EUR). The polygenic score for cardiomyopathy (PGS002051) has been derived in the UK Biobank where it has an inflated estimate of effect size making a comparison to Arabs inaccurate. N total is the total number of samples in the validation dataset

excluding missing values for each disease. Odds ratio per standard deviation of the score is derived from a logistic regression model adjusted for age, sex, array version, and first 10 principal components of ancestry. The black boxes indicate the adjusted odds ratio. The horizontal lines around the black boxes indicate the 95% confidence intervals. **d** Performance of polygenic scores for LDL cholesterol (LDL-C), HDL cholesterol (HDL-C), triglycerides (TG), systolic blood pressure (SBP), diastolic blood pressure (DBP), body mass index (BMI), and height in Arabs from this study (dark green) vs. a matched case-control sample of European-ancestry individuals from the UK Biobank (EUR, dark red). Sample sizes for all traits are included in Supplementary Table 2. Effect size estimates are derived from linear regression models adjusted for age, sex, array version, and first 10 principal components of ancestry. Error bars represent the standard error.

**Table 1 | Characteristics of study participants**

| Characteristic | Case-control cohort (N = 5399) |
|---|---|
| Male sex, n (%) | 3473 (64.3) |
| Age, mean (SD) | 54.83 (14.82) |
| Coronary artery disease, n (%) | 3491 (65.0) |
| Type 2 diabetes, n (%) | 2979 (55.6) |
| Cardiomyopathy, n (%) | 479 (9.3) |
| LDL cholesterol[a], mean (SD) [mg/dL] | 135.82 (53.38) |
| HDL cholesterol[b], mean (SD) [mg/dL] | 45.20 (13.24) |
| Triglycerides[c], mean (SD) [mg/dL] | 165.73 (111.87) |
| Systolic blood pressure[d], mean (SD) [mmHg] | 146.61 (23.92) |
| Diastolic blood pressure[d], mean (SD) [mmHg] | 83.06 (13.34) |
| Body mass index, mean (SD) [kg/m²] | 29.31 (6.05) |
| Height, mean (SD) [m] | 1.61 (0.09) |
| Current or former smoking, n (%) | 2060 (38.3) |

The number of missing values varies across the characteristics.
[a]LDL cholesterol levels were adjusted for statin, ezetimibe, and fibrate use.
[b]HDL cholesterol levels were adjusted for fibrate use.
[c]Triglycerides levels were adjusted for statin and fibrate use.
[d]Systolic and diastolic blood pressures were adjusted for antihypertensive medication (ACE inhibitors, angiotensin receptor blocker, beta blockers, calcium channel blockers, diuretics, and nitrates) use.

decrease in performance for CAD, type 2 diabetes, and the continuous traits (Fig. 2c, d). Comparing European to Arab, the odds ratio per standard deviation (OR per SD) for CAD was 1.78 (95% CI 1.69-1.88) vs. 1.41 (95% CI 1.31–1.50), and for type 2 diabetes was 1.82 (95% CI 1.73-1.91) vs. 1.41 (95% CI 1.32–1.49) (Fig. 2c). The seven cardiometabolic traits had a mean decrease of performance of 42.9% (95% CI 4.4–81.3), measured as difference in the effect size of the score on the trait obtained from a linear regression model (Fig. 2d).

### Arab-optimized polygenic scores on par with European scores

After establishing that currently used polygenic scores have reduced performance in Arabs, we sought to follow the pragmatic approach to optimize polygenic scores using entirely public datasets (Fig. 1). For each trait, we obtained variant effect size from the largest and most diverse GWAS available (Supplementary Table 3) and computed scores using five methods—PRSice-2, LDpred2, lassosum2, PRS-CS, and PRS-CSx (Supplementary Table 4). Scores were adjusted for principal components of ancestry as described previously[7,35].

In the training set (N = 2700), we identified the best performing score, defined as having the largest area under the curve (AUC) in a logistic regression model for binary disease classifications or having the largest adjusted coefficient of determination (adjusted $R^2$) in a linear regression model for continuous traits (Supplementary Tables 6 and 7). The best-performing score was then tested in a validation set (N = 2699) (Supplementary Table 8). The number of single nucleotide polymorphisms for the best-performing scores ranged from 10,440 to 1,069,677 depending on the trait (Table 2). There was no single scoring method that stood out as superior across all traits, but methods using Bayesian shrinkage or Lasso regression that factor genetic architecture had higher performance compared to a baseline method such as PRSice-2 (clumping and thresholding)[31–34]. Similarly, factoring LD structure using the new method PRS-CSx performed favorably (Table 2)[6].

Three Arab-optimized scores for CAD, type 2 diabetes, and cardiomyopathy had marked improvement in performance compared to a publicly available European score performance in Arabs—OR per SD was 1.51 (95% CI 1.42–1.61) for CAD, 1.83 (95% CI 1.74–1.92) for type 2 diabetes, and 1.34 (95% CI 1.13–1.64) for cardiomyopathy (Table 2).

Similarly for continuous traits, Arab-optimized scores had an augmented performance compared to a European score for 5 out of 7 traits (Table 2). More importantly, for some traits the Arab-optimized score performance was on par with known performance of published scores in individuals of European ancestry suggesting that even within the limitation of absence of large GWAS and imputation servers for Arabs, using the pragmatic approach we described might enable equitable implementation. For example, the optimized score for BMI derived using PRS-CSx had an effect size of 1.18 kg/m² per SD and adjusted $R^2$ of 0.09, a notable augmentation from an effect size of 0.97 kg/m² per SD and adjusted $R^2$ of 0.11 for the the performance of the published European-derived score in Arabs (Table 2)[39]. For some traits, the augmentation was minimal such as for LDL cholesterol, whereby the published score is already known to be highly performing owing to its development from massive GWAS datasets[40]. Notably for certain traits, performance was still lower than reported in individuals of European ancestry such as CAD where the OR per SD was 1.51 compared to estimates above 1.6 for prevalent disease in most cohorts[2,41,42].

Finally, we tested how imputation of genotyping data might affect performance of optimized scores. Scores derived using genotyping array data without imputation had a reduced performance compared to with imputation, but scores derived from data imputed on TOPMed vs. 1000 G panels did not vary significantly (Supplementary Fig. 2)[43,44].

### Clinical utility of polygenic scores for cardiometabolic disease in Arabs

The clinical utility of polygenic scores for cardiometabolic disease stems from their ability to stratify disease risk in the population. We defined percentiles of polygenic risk based on the reference population (Supplementary Fig. 1) and confirmed that the distribution of scores in the control is comparable to their distribution in the reference population (Supplementary Fig. 3). In the Arab case-control cohort, we see a marked stratification of disease risk by polygenic scores. The prevalence of CAD ranged from 43.7% in the bottom decile to 74.4% in the top decile of the polygenic score. Individuals in the top decile of polygenic score had 4-fold increased risk of CAD (OR 3.94, 95% CI 3.51–4.36) compared to individuals in the bottom decile (Fig. 3a). The prevalence of type 2 diabetes ranged from 31.9% in the bottom decile to 74.9% in the top decile—odds ratio of 8.19 (95% CI 7.76–8.61) comparing top to bottom decile (Fig. 3b). There were also significant differences in continuous traits by polygenic score—LDL cholesterol was 108.28 (IQR 85.07–143.36) mg/dL in the bottom decile compared to 146.09 (IQR 110.49–182.30) mg/dL in the top decile of polygenic score (p-value = 8.49e-13), and BMI was 26.30 (IQR 23.30–30.30) kg/m² in the bottom decile compared to 31.20 (IQR 27.07-35.48) kg/m² in the top decile of polygenic score (p-value = 6.39e-16) (Fig. 3c, d). Similar stratification of risk and measured traits was seen for cardiomyopathy, HDL cholesterol, triglycerides, systolic blood pressure, diastolic blood pressure and height (Supplementary Fig. 4).

### Interplay of polygenic scores with conventional risk factors

We then asked whether polygenic scores have additive effects to conventional risk factors which is particularly important in an Arab population with high prevalence of those risk factors. In the study population, 2060 (38.3%) were current or former smokers, 2126 (41.5%) were obese, 4205 (80.6%) had hypertension, 2979 (55.6%) had type 2 diabetes, and 1783 (40.1%) had hypercholesterolemia (Table 1). The effect of polygenic score on CAD remained unchanged after adjusting for smoking, obesity, systolic blood pressure, type 2 diabetes, and LDL cholesterol (Supplementary Table 8). There was also no interaction between each of these risk factors and the polygenic score for CAD (p-values for interactions > 0.12). For type 2 diabetes, the effect of polygenic score was slightly higher after adjusting for BMI—OR per SD was

**Table 2 | Performance of optimized polygenic scores for cardiometabolic traits in Arabs**

| Traits | European-derived polygenic scores | | | | | Arab optimized polygenic scores | | | |
|---|---|---|---|---|---|---|---|---|---|
| **Categorical traits** | **PGS Catalog ID** | **Derivation Method** | **No. of variants in PRS** | **OR per SD (95% CI)** | **AUC (95% CI)** | **Derivation Method** | **No. of variants in PRS** | **OR per SD (95% CI)** | **AUC (95% CI)** |
| Coronary artery disease | PGS000013 | LDpred | 5,706,928 | 1.41 (1.31–1.50) | 0.7909 (0.7726–0.8091) | lassosum2 | 10,440 | 1.51 (1.42–1.61) | 0.7950 (0.7768–0.8132) |
| Type 2 diabetes | PGS000014 | LDpred | 5,786,938 | 1.41 (1.32–1.49) | 0.7054 (0.6854–0.7255) | PRS-CS | 1,068,166 | 1.83 (1.74–1.92) | 0.7384 (0.7194–0.7574) |
| Cardiomyopathy | PGS002051 | LDpred2 | 621,802 | 1.01 (0.88–1.16) | 0.6277 (0.5890–0.6665) | LDpred2 | 1,010,014 | 1.34 (1.13–1.64) | 0.6453 (0.6086–0.6819) |
| **Continuous traits** | **PGS Catalog ID** | **Derivation Method** | **No. of variants in PRS** | **Effect size per SD (SE)** | **Adjusted R-sq** | **Derivation Method** | **No. of variants in PRS** | **Effect size per SD (SE)** | **Adjusted R-sq** |
| LDL Cholesterol [mg/dL] | PGS000892 | PRS-CS | 1,068,974 | 10.06 (1.10) | 0.0405 | PRS-CSx | 1,069,677 | 9.40 (1.10) | 0.0358 |
| HDL Cholesterol [mg/dL] | PGS002781 | PRS-CS | 1,112,500 | 3.49 (0.27) | 0.1351 | PRS-CSx | 1,069,677 | 3.67 (0.27) | 0.1424 |
| Triglyceride [mg/dL] | PGS002784 | Pruning and Thresholding | 12,709 | 25.19 (2.34) | 0.0682 | lassosum2 | 54,623 | 29.32 (2.31) | 0.0857 |
| Systolic blood pressure [mmHg] | PGS002238 | PRS-CS | 1,069,203 | 1.08 (0.45) | 0.0961 | PRS-CS | 1,056,790 | 3.10 (0.44) | 0.1108 |
| Diastolic blood pressure [mmHg] | PGS002239 | PRS-CS | 1,069,056 | 0.71 (0.26) | 0.0249 | lassosum2 | 25,857 | 1.80 (0.26) | 0.0397 |
| Body mass index [kg/m²] | PGS000027 | LDpred | 2,015,065 | 0.97 (0.11) | 0.0790 | PRS-CSx | 1,067,771 | 1.18 (0.11) | 0.0919 |
| Height [m] | PGS002804 | SBayesC | 1,054,056 | 0.027 (0.0013) | 0.5299 | PRS-CSx | 1,067,771 | 0.026 (0.0013) | 0.5202 |

Performance of European-derived polygenic scores and Arab-optimized polygenic scores for each trait are shown in the validation dataset of Arabs ($N = 2699$). European-derived polygenic scores were calculated using scoring files from The Polygenic Score (PGS) Catalog with datasets consisting primarily of European ancestry and not derived from the UK Biobank. Further information about the scoring files, including their GWAS sources, is included in Supplementary Table 1. Arab-optimized polygenic scores were derived in this study. The number of missing values varies across the disease or trait. Additional details are shown in Supplementary Table 5. OR per SD and AUC were determined using a logistic regression model adjusted for age, sex, array version, and the first 10 principal components of ancestry. Effect size per SD and adjusted $R^2$ values for continuous traits were determined using a linear regression model with similar covariates. (*OR per SD* odd ratio per standard deviation, *AUC* area under the receiver operating characteristic curve, *CI* confidence interval, *SE* standard error).

2.36 (95% CI 1.86-2.85) in BMI-adjusted model compared to 1.83 (95% CI 1.77–1.90) in unadjusted model, but the interaction term was not statistically significant (*p*-value for interaction = 0.39).

We then evaluated whether the effect of the polygenic score is consistent across different demographic and clinical subgroups of patients. Genomic risk was strongly associated with the severity of CAD as evaluated on coronary angiogram. For patients with all three major coronary arteries with obstructive disease ("three-vessel CAD"), the OR per SD was 1.78 (95% CI 1.66–1.90) compared to an OR per SD of 1.41 (95% CI 1.29–1.53) for patients with only one obstructive vessel ("one-vessel CAD") (Fig. 4). There were variations in the effect size of the polygenic score by age of CAD, sex, and presence of diabetes, but those were not statistically significant (*p*-value of interaction terms > 0.05) (Fig. 4). For type 2 diabetes, the effect of the polygenic score was also consistent across different demographic and clinical subgroups of patients (Supplementary Fig. 5).

Finally, we compared the characteristics of CAD and type 2 diabetes patients with high (top quintile) vs. low (bottom quintile) polygenic risk (Supplementary Tables 9 and 10). CAD patients with high polygenic risk ($N = 1042$) had earlier onset (58.1 vs. 60.3 years, $p = 0.001$) and more severe disease (37.1% vs. 24.1% with three-vessel CAD, $p = 8.5e$-09) despite similar clinical risk factors, compared to CAD patients with low polygenic risk ($N = 441$). Among 1042 patients with CAD and high polygenic risk, there were 545 (52.2%) non-smokers, 636 (58.9%) non-obese, 189 (14.9%) with no hypertension, 338 (32.4%) with no diabetes, and 626 (53.3%) with normal cholesterol. Type 2 diabetes patients with high polygenic risk ($N = 1008$) had earlier onset of diabetes (58.5 years vs. 61.0 years, $p = 5.2e$-04) and were more likely to be

on insulin therapy (42.6% vs. 30.5%, $p = 1.2e$-04) compared to type 2 diabetes patients with low polygenic risk ($N = 321$).

## Identifying and validating polygenic scores in Arabs living in the UK

Given the scarcity of Arab cohorts, we apply an approach that uses principal components of ancestry from the indigenous Arab cohort of Saudi Arabia to identify an ancestry-matched cohort in the UK Biobank (Methods, Supplementary Fig. 6). Using a strict definition for genetic distance, we identified 420 "Arab-matched" participants in the UK Biobank based only on genetic information. Of note, a self-report of "Arab" or "Middle Eastern" is not a captured ethnicity in the UK Biobank. Out of the 420 participants we identified based on genetic distance, 365 or 86.9% reported being of "other ethnic group" or "any other White/Asian background" (Supplementary Fig. 6).

We tested the performance of both European-derived scores and our Arab-optimized scores for continuous traits in this "Arab-matched" cohort in the UK Biobank compared to their performance in 1000 randomly selected groups of European ancestry from the UK Biobank. European-derived scores had 27% reduced performance in the Arab-matched cohort compared to the European-ancestry cohort. The optimized scores from our study improved performance among both Arab-matched and European ancestry participants in the UK Biobank for systolic blood pressure, diastolic blood pressure and triglycerides, but did not result in a significant improvement in performance for the other traits (Supplementary Fig. 7). Heterogeneity of effect due to polygenic score selection used for comparison and small sample size of the Arab-matched cohort is a limitation of this analysis.

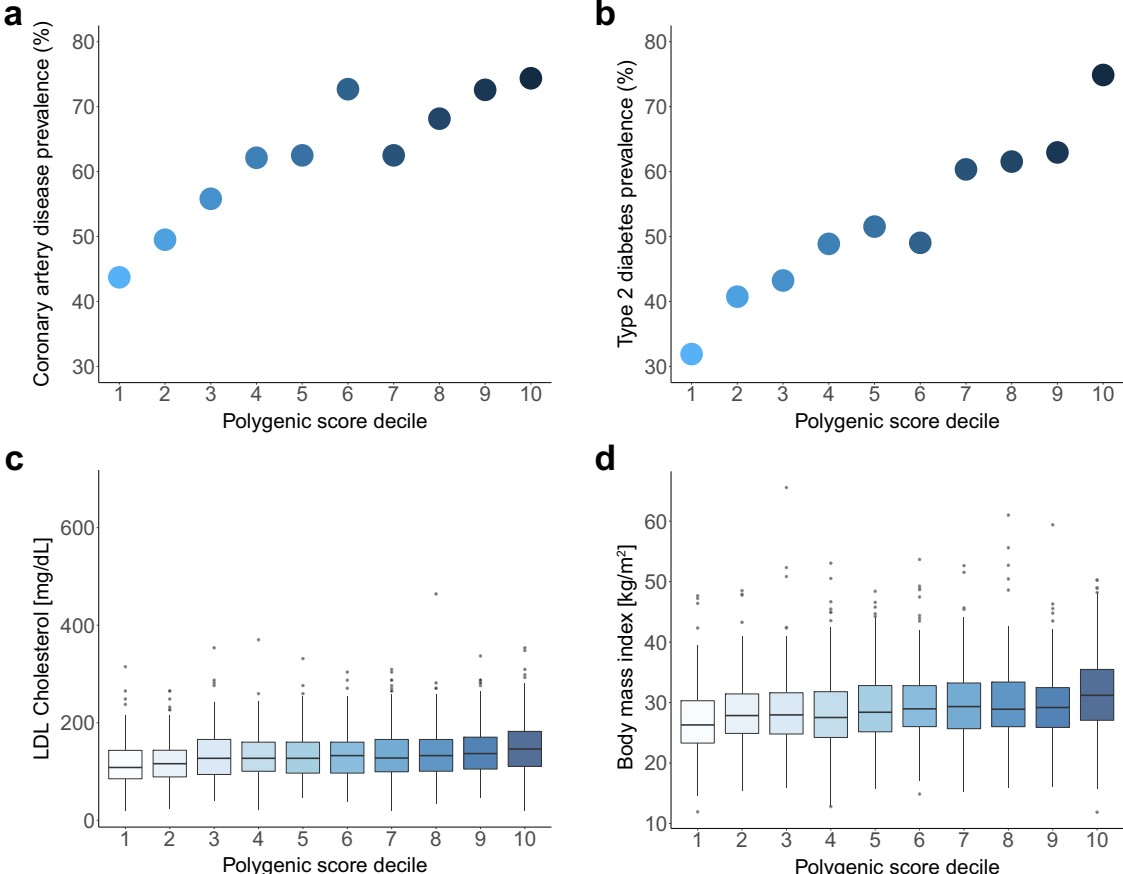

**Fig. 3 | Stratification of cardiometabolic disease by polygenic scores among Arabs.** **a**, **b** Prevalence of coronary artery disease and type 2 diabetes in the validation dataset based on polygenic score decile rank derived from the reference dataset. **c**, **d** Distribution of LDL cholesterol and body mass index in the validation dataset based on polygenic score decile groups. After excluding missing values, the sample sizes were 2201 for LDL cholesterol and 2553 for body mass index. The horizontal lines within each boxplot represent the median, the top, and bottom of each box indicate the interquartile range, and the whiskers reflect the maximum and minimum values within each group.

## Discussion

We describe a pragmatic approach for optimizing polygenic scores for 10 cardiometabolic traits in Arabs and show a strong promise for clinical utility. This framework leverages publicly available resources—short of much-needed large GWAS in non-Europeans—and could be generalizable to other populations to advance equitable clinical implementation of polygenic scores.

Our findings have three key implications. First, we illustrate how public resources could be leveraged to study the clinical utility of polygenic scores in a subpopulation that is distant genetically and geographically from global datasets (such as indigenous Arabs of Saudi Arabia), and validate them in ancestry-matched individuals in a European biobank. The PGS catalog has been an instrumental tool in making scores available, and variant effect size is becoming increasingly available from large multi-ancestry GWAS[38]. While those are disproportionately enriched for European ancestries, for certain cardiometabolic traits they might still include large absolute numbers of individuals of non-European ancestries. For example a recent lipid GWAS included 334,944 individuals of non-European ancestries and a multi-ancestry polygenic score for LDL cholesterol reported in that study performed favorably across 6 ancestries[40]. Methods to compute scores are also improving and becoming more accessible through online tutorials and user-friendly software[31,45]. While increasing representation in genomic datasets should remain a priority, we demonstrate that existing resources could be helpful in cross-ancestry implementation of polygenic scores in the case of Arabs.

Second, we show that polygenic risk is additive to conventional risk factors even in a population with very high prevalence of cardiometabolic disease. Saudi Arabia, similar to several Arab Gulf countries, has an alarming prevalence of obesity, type 2 diabetes, and other related cardiometabolic disease[17,19,20]. For example, the prevalence of type 2 diabetes exceeds 20% and has increased by 95% between 2009 and 2019[16]. Polygenic scores powerfully stratified disease in this population, were independent of conventional risk factors, identified a significant proportion of the population with high genomic risk without conventional risk factors, and correlated with disease severity in clinically meaningful ways. We note that despite a lower relative risk of disease associated with a score in Arabs compared to Europeans—odds ratio per standard deviation for CAD is -1.51 compared to >1.60—the net benefit in terms of identifying more individuals at risk might still be high in a population with high prevalence of disease. This has been shown among Black individuals in the USA in a recent study whereby the polygenic score improved the estimation of absolute risk of myocardial infarction in Black individuals more than in White individuals despite a weaker association of the score with disease[3]. The particularly strong performance of the polygenic score for body mass index is noteworthy because the very high rate of obesity among Saudis (half the population) has been largely attributed to lifestyle changes[17]. The rapid transition from a nomadic lifestyle with scarce food to sedentary lifestyle with food abundance is a phenomenon that has been observed in other ethnicities with exceptionally high

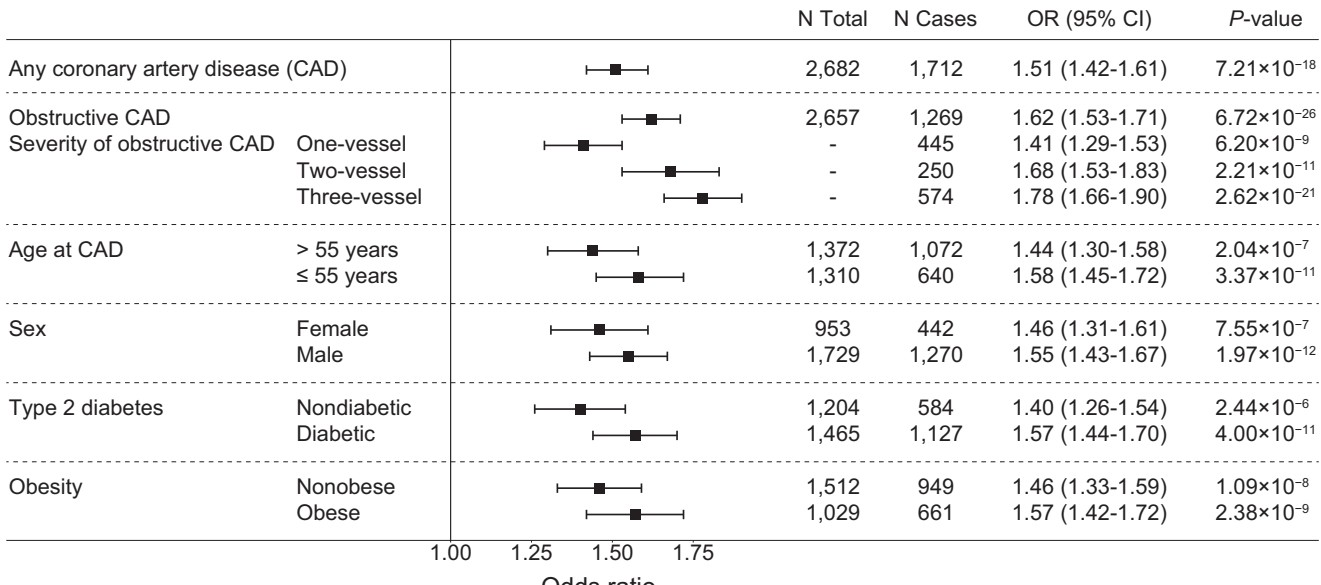

| | | N Total | N Cases | OR (95% CI) | P-value |
|---|---|---|---|---|---|
| Any coronary artery disease (CAD) | | 2,682 | 1,712 | 1.51 (1.42–1.61) | 7.21×10⁻¹⁸ |
| Obstructive CAD | | 2,657 | 1,269 | 1.62 (1.53–1.71) | 6.72×10⁻²⁶ |
| Severity of obstructive CAD | One-vessel | - | 445 | 1.41 (1.29–1.53) | 6.20×10⁻⁹ |
| | Two-vessel | - | 250 | 1.68 (1.53–1.83) | 2.21×10⁻¹¹ |
| | Three-vessel | - | 574 | 1.78 (1.66–1.90) | 2.62×10⁻²¹ |
| Age at CAD | > 55 years | 1,372 | 1,072 | 1.44 (1.30–1.58) | 2.04×10⁻⁷ |
| | ≤ 55 years | 1,310 | 640 | 1.58 (1.45–1.72) | 3.37×10⁻¹¹ |
| Sex | Female | 953 | 442 | 1.46 (1.31–1.61) | 7.55×10⁻⁷ |
| | Male | 1,729 | 1,270 | 1.55 (1.43–1.67) | 1.97×10⁻¹² |
| Type 2 diabetes | Nondiabetic | 1,204 | 584 | 1.40 (1.26–1.54) | 2.44×10⁻⁶ |
| | Diabetic | 1,465 | 1,127 | 1.57 (1.44–1.70) | 4.00×10⁻¹¹ |
| Obesity | Nonobese | 1,512 | 949 | 1.46 (1.33–1.59) | 1.09×10⁻⁸ |
| | Obese | 1,029 | 661 | 1.57 (1.42–1.72) | 2.38×10⁻⁹ |

Odds ratio: 1.00  1.25  1.50  1.75

**Fig. 4 | Performance of coronary artery disease polygenic score in different patient subgroups.** In the validation dataset, the effect of the coronary artery disease (CAD) polygenic score in various clinical subgroups of patients is evaluated. The association of the score with CAD increased with severity of disease as defined by invasive coronary angiography. Obstructive CAD is defined as luminal stenosis of at least 50% in the left main coronary artery or at least 70% in any other coronary artery. The number of vessels with obstructive CAD indicates an increasing burden/severity of disease and manifested a stronger association with the polygenic score. The association of the polygenic score with CAD was consistent across multiple clinical and demographic subgroups. Obesity was defined as body mass index ≥ 30 kg/m². The odds ratio was assessed in a logistic regression model with age, sex, array version, and the first 10 principal components of ancestry as covariates. For the age and sex subgroups, we excluded those variables from the covariates. P-values were determined using a two-sided Wald test. The black boxes indicate the adjusted odds ratio. The horizontal lines around the black boxes indicate the 95% confidence intervals.

rates of obesity such as Native Americans[46]. Our finding suggests that this lifestyle transition-induced obesity has been influenced by genetic risk factors that varied in their distribution in the local population.

Third, we live in a global world and self-reported race and ethnicity is increasingly proving to be an inappropriate form of health-related measures[47–49]. Genetic ancestry in the case of polygenic scores might be a more appropriate measure to advance cross-ancestry implementation. In this study, we identify Arab-matched individuals in the UK Biobank who self-report their race to be a variety of options, yet have similar genetic ancestry to Arabs from Saudi Arabia and could benefit from optimized polygenic scores for this population rather than European polygenic scores. We propose similar approaches be applied across traits and populations to advance cross-ancestry implementation of polygenic scores without being limited by a small number of continental populations.

The study has few limitations that are important to note. First, this is cross-sectional case-control study without longitudinal follow-up, which limits evaluation of risk longitudinally and population estimates of clinico-genomic risk. Second, this study is focused on indigenous Arabs and does not cover Levantine or North African Arabs, although we note that the study population is closer to those populations than Europeans suggesting that our findings are relevant especially as we think of genetic distance as a continuous measure[50]. Third, factoring LD structure using a method such as PRS-CSx did improve performance, which raises the question whether imputing on an Arab imputation panel could improve polygenic score performance, but those are not yet publicly available. Fourth, phenotype quality in this study was obtained in the context of routine care and as such was of variable quality depending on the trait. For example, there was a uniquely detailed and accurate CAD phenotype from cardiac catheterization data, but cardiomyopathy phenotype was based on mention in a clinical note by a physician without additional details.

## Methods

### Study populations

The study population included a disease cohort of 5399 individuals referred for cardiology care at the King Faisal Specialist Hospital and Research Center—a tertiary care hospital in Riyadh—from all five regions of Saudi Arabia, and a population reference cohort of 1017 individuals not known to have cardiometabolic disease and that are representative of 28 tribes of indigenous Arabs in Saudi Arabia[27,36]. Participants in the disease cohort additionally provided access to the electronic health records to obtain phenotype data and provided additional socio-demographic background information obtained by a clinical research coordinator. All participants provided blood samples for DNA extraction and genotyping array analyses and informed consent to participate in the study. The study was approved by the institutional review board (KFSHRC RAC# 2190011). This study was established through a partnership with Saudi Arabia and included capacity-building for advancing polygenic score research and implementation.

### Phenotype definitions

Disease status and measured cardiometabolic traits were all collected in the context of clinical care and extracted manually from the electronic health record by trained medical personnel. Coronary artery disease was defined as either presence of any coronary atherosclerosis on cardiac catheterization or a diagnosis of myocardial infarction. Obstructive coronary artery disease was defined as at least 70% stenosis in any of the left anterior descending, left circumflex, or right coronary arteries, or at least 50% in the left main coronary artery. Coronary artery disease severity was classified by the number of vessels that have obstructive coronary artery disease (1, 2 or 3). Diagnoses of type 2 diabetes and cardiomyopathy as well as detailed medication history were curated from the electronic health records. Lipid levels were measured in the context of clinical care and the first lipid level measured on referral to the King Faisal Specialist Hospital and

Research Center was used. Lipid levels were adjusted for lipid-lowering medication intake to estimate untreated lipid levels[51]. Briefly, we estimated that statins reduce LDL cholesterol by 30% and triglycerides by 15%. In addition, we assumed that ezetimibe and fibrate lowered LDL cholesterol by 20% and 10%, respectively, if used separately. Similarly, blood pressure measurements, and height and weight measurements obtained in the context of care during the first visit were used in the study. Blood pressure was also adjusted for anti-hypertensive medication intake by adding 15 mmHg to the systolic blood pressure and 10 mmHg to the diastolic blood pressure[52,53].

### Genotyping quality control and imputation
Genotyping was performed using Affymetrix Axiom Genome-Wide ASI Array and array versions na29 and na32 as described in a prior study[27]. Out of 6566 samples, quality control excluded 150 samples due to variant calling missingness > 5%, heterozygosity rate > 5 standard deviations above the mean, and mismatch between genotypically-determined and self-reported sex, resulting in 6416 samples consisting of 5399 patients and 1017 reference samples. Variant-level quality control was conducted to remove variants with call rate < 98%, minor allele frequency (MAF) < 0.01, and Hardy–Weinberg equilibrium $P < 1 \times 10^{-6}$. We performed imputation using both 1000 Genomes Project Phase 3 and TOPMed version R2 datasets[43,44]. When the results were compared, there was no significant difference (Supplementary Fig. 2), hence downstream analyses were performed with imputed data using 1000 Genomes Project Phase 3 dataset as a reference panel. After imputation, the variants with imputation quality scores (INFO) < 0.3 and MAF < 0.01 were removed resulting in 8,469,565 variants.

### Principal components of ancestry
A static genetic ancestry reference distribution of Arabs was generated by principal component analysis (PCA) using FlashPCA software[54]. A set of variants from the intersection between the Arab genotype data and the 1000 Genomes Project Phase 3 was used to perform PCA. After doing quality control on 2481 unrelated individuals from the 1000 Genomes Project Phase 3 (--chr 1-22 --snps-only --maf 0.01 --geno 0.02 --hwe 1e-6 --mind 0.05), we extracted the overlapping variants and pruned for linkage disequilibrium (--indep-pairwise 1000 50 0.2) using PLINK 2.0, yielding 177,917 variants[55]. Then, we performed PCA using the pruned variants in 1000 Genomes Project Phase 3 and projected Arab individuals onto the created principal component space.

### Polygenic score calculation
To examine the performance of European-derived polygenic scores for three diseases and seven traits, we downloaded scoring files from the PGS catalog and computed the scores using PLINK 2.0 (Supplementary Table 1)[38]. We prioritized scores that are not derived from the UK Biobank to enable comparison of performance of those scores in Arabs to individuals of European ancestry from the UK Biobank. For each of the 10 traits, we derived a matched cohort of equal size from individuals of European ancestry in the UK Biobank using the R package "MatchIt"[56]. Briefly, this technique uses 1:1 nearest neighbor matching to generate an equal sized cohort that is as similar to the study population as possible in terms of age, sex, and case-control ratio for categorical traits, or mean value for continuous traits.

For Arab-optimized polygenic scores, we selected those with a large sample size and diverse ethnicities from currently accessible genome-wide association study (GWAS) results (Supplementary Table 3). The effect sizes of variants from the GWAS results were chosen or adjusted to calculate the raw polygenic scores using five latest approaches including PRSice-2, LDpred2, lassosum2, PRS-CS, and PRS-CSx (Supplementary Table 4)[6,31–34]. We utilized an in-sample LD matrix for PRSice-2, and external LD reference panels constructed using the 1000 Genomes Project Phase 3 dataset for the remaining approaches. For LDpred2, lassosum2, and PRS-CS, we used an LD

reference panel built using European samples ($N = 503$). For PRS-CSx, we used LD reference panels created from the Arab population of this study ($N = 6416$) and five super populations in the 1000 Genomes Project. The ancestry-adjusted scores were obtained by adjusting the first 10 principal components in a linear regression model and taking the difference between the raw and the predicted scores as previously described[7,35]. To facilitate interpretation, all scores were normalized.

### Identifying best-performing polygenic score
To determine the optimal polygenic score model, the study population was split into training and validation datasets. Using --thin-indiv-count < n > in PLINK 2.0, which randomly removes samples until n samples remain, we defined $n = 2700$. This resulted in a training dataset of 2700 samples. The remaining 2699 samples were utilized as a validation dataset (Supplementary Table 5). Maximum area-under-curve (AUC) or adjusted $R^2$ of the score evaluated in a regression model with age, sex, array version, and first 10 principal components in a training set was used to determine the best model for each trait (Supplementary Tables 6 and 7).

### Defining polygenic score percentiles
To generate a polygenic score reference distribution, 1017 Arabs from the 28 tribes of Saudi Arabia were selected without consideration of any disorders[36]. This reference population had similar genetic ancestry to the case-control study population (Supplementary Fig. 1), and the distribution of polygenic scores in the reference population was similar to the control (Supplementary Fig. 3). Polygenic risk of individuals in a disease cohort was evaluated relative to a reference population using a linear regression model based on the reference dataset. Residual scores generated from the first 10 principal components were used to estimate the mean and standard deviation of the reference distribution. The predict function in R was then applied with the regression model from the reference dataset to derive adjusted scores for individuals in the disease cohort. The adjusted scores were further standardized using the mean and standard deviation of the reference distribution to obtain the percentile ranks[7]. The percentile distribution of polygenic scores was assessed for stratification of 10 cardiometabolic diseases and traits.

### Studying the interplay of polygenic score with conventional risk factors
For each polygenic score, we grouped the disease cohort in deciles of that score defined using the reference population. In each decile, we identified the proportion of participants with the disease of interest (e.g. coronary artery disease) or median of the trait (e.g. LDL cholesterol). To test the interplay of conventional risk factors with CAD and type 2 diabetes, we used logistic regression models that include conventional risk factors, individually, in groups, and as interaction terms with the polygenic score. We also compared the characteristics of individuals with high vs. low genomic risk, defined as top and bottom quintiles of the population distribution of polygenic scores respectively using Welch's or Student's two sample t-test.

### Identifying Arab-matched Individuals in the UK Biobank
The UK Biobank is a prospective national biobank study that enrolled about half a million middle-aged adult participants between 2006 and 2010 and has detailed phenotypic data and whole-genome genotyping data imputed centrally[37]. Analysis of the UK Biobank data was performed using application 31224 and approved by KAUST IRB. We used principal components of ancestry to identify Arab-matched individuals in the UK Biobank by defining a genotypic distance from the centroid of our Arab samples from Saudi Arabia. To this end, we identified 51,916 common SNPs among the UK Biobank ($N = 223,901$), unrelated Arab samples ($N = 5884$), and 1000 Genomes Project samples ($N = 2504$), performed PCA on the 1000 Genomes Project samples, and

projected the UK Biobank and the unrelated Arab samples onto a subspace spanned by the top 10 principal components of the 1000 Genomes Project samples. We then computed the squared Mahalanobis distance of the UK Biobank samples from the centroid of the unrelated Arab samples. To find a boundary cutoff to determine Arab-matched samples, we used the critical value from the chi-squared test at significance level 0.05 with 10 degrees of freedom.

For Arab-matched individuals ($N = 420$), we evaluated the distribution of self-reported ancestry. Participants in the UK Biobank answered the question "What is your ethnic group?" which had the following options: "White" ("British", "Irish", or "Any other white background"), "Mixed" ("White and Black Caribbean", "White and Black African", "White and Asian", or "Any other mixed background"), "Asian or Asian British" ("Indian", "Pakistani", "Bangladeshi", or "Any other Asian background"), "Black of Black British" ("Caribbean", "African", or "Any other Black background"), "Chinese", "Other Ethnic Group", "Do not know", or "Prefer not to answer". Notably, there were no options for Middle Eastern or Arab.

To compare performance of scores in Arab-matched samples to individuals of European ancestry, we defined a European ancestry cohort using a genetically defined Caucasian ethnic group reported in the UK Biobank (field 22006) ($N = 223,901$). For both Arab-matched participants ($N = 420$) and European ancestry cohort ($N = 223,901$) we obtained phenotypic data for body mass index (field 21001), height (field 50), LDL cholesterol (field 30780), HDL cholesterol (field 30760), triglycerides (field 30870), systolic blood pressure (field 4080), diastolic blood pressure (field 4079) and medication intake history (field 6177) were obtained from the UK Biobank (Supplementary Table 11). Lipid levels and blood pressure were also adjusted for medication intake as described in the phenotype definition section. For this comparison, we randomly selected 1000 subsets of 420 samples from the European ancestry cohort, computed scores for the Arab-matched and 1000 European ancestry groups for each trait, and calculated effect size per SD and adjusted $R^2$ as performance measures.

**Reporting summary**

Further information on research design is available in the Nature Portfolio Reporting Summary linked to this article.

## Data availability

Due to local privacy laws and privileged human information, all requests for raw genotyping and clinical data are subject to prior approval from the local IRB. For the raw data for the Arab cohort, the local IRB can be reached at ORA@kfshrc.edu.sa with an expected timeframe for response of 2 months. Analysis of the UK Biobank data was performed using application 31224 and approved by King Abdullah University of Science and Technology (KAUST) IRB. The UK Biobank data are available to researchers with research inquiries following IRB and UK Biobank approval (https://www.ukbiobank.ac.uk/enable-your-research/apply-for-access). The GWAS Catalog (https://www.ebi.ac.uk/gwas/downloads/summary-statistics) contains all GWAS summary statistics. Ancestry-matched LD reference panels built with 1000 Genomes Project phase 3 samples are available at https://github.com/getian107/PRScsx. The polygenic scores described in this publication are available for download from the Polygenic Score Catalog (https://www.pgscatalog.org) under the publication ID PGP000501 and the score IDs PGS003866-PGS003891.

## Code availability

The manuscript described a pragmatic approach of publicly available software. Additional code is available upon request from the authors.

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

## Acknowledgements

This study was supported by a sponsored research agreement between King Faisal Specialist Hospital and Research Center and Massachusetts General Hospital. I.S. is supported by a grant of the Korea Health Technology R&D Project through the Korea Health Industry Development Institute (KHIDI), funded by the Ministry of Health & Welfare, South Korea (HI19C1328). H.K., N.C. and X.G. are supported by the King Abdullah University of Science and Technology (KAUST) Office of Research Administration (ORA) under Award No FCC/1/1976-44-01, FCC/1/1976-45-01, REI/1/5202-01-01, REI/1/5234-01-01, REI/1/4940-01-01, RGC/3/4816-01-01, and REI/1/0018-01-01. H.H.W. is supported by the National Research Foundation of Korea of Korea Grant funded by the Ministry of Science and Information and Communication Technologies, South Korea (NRF-2022R1A2C2009998). P.N. is supported by grants from the National Heart, Lung, and Blood Institute (R01HL148050, R01HL151283, R01HL148565), National Human Genome Research Institute (U01HG011719) and Fondation Leducq (TNE-18CVD04). P.T.E. is supported by grants from the National Institutes of Health (1RO1HL092577, 1R01HL157635, 5R01HL139731), from the American Heart Association Strategically Focused Research Networks (18SFRN34110082), and from the European Union (MAESTRIA 965286). A.C.F. is supported by grants from the National Heart, Lung, and Blood Institute (1K08HL161448 and R01HL164629).

## Author contributions

I.S., A.V.K., F.S.A., and A.C.F. conceived and designed the study. I.S., H.K., N.C., M.O.H., L.A., M.A., X.G., F.S.A., and A.C.F. acquired, analyzed, and interpreted the data. I.S., F.S.A., and A.C.F. drafted the manuscript.

H.K., N.C., M.O.H., L.A., M.A., H-H.W., P.N., P.T.E., A.V.K., and X.G. critically revised the manuscript for important intellectual content.

## Competing interests

P.N. reports grant support from Amgen, Apple, AstraZeneca, Boston Scientific, and Novartis, personal fees from Allelica, Apple, AstraZeneca, Blackstone Life Sciences, Foresite Labs, Genentech/Roche, and Novartis, is a scientific advisory board member of Esperion Therapeutics, geneXwell, and TenSixteen Bio, is a co-founder of TenSixteen Bio, and spousal employment at Vertex, all unrelated to the present work. P.T.E. has received sponsored research support from Bayer AG and IBM Health, and he has served on advisory boards or consulted for Bayer AG, MyoKardia and Novartis. A.V.K. is an employee of Verve Therapeutics; has served as a scientific advisor to Amgen, Novartis, Silence Therapeutics, Korro Bio, Veritas International, Color Health, Third Rock Ventures, Illumina, Ambry, and Foresite Labs; holds equity in Verve Therapeutics, Color Health, and Foresite Labs; and is listed as a co-inventor on patent applications related to assessment and mitigation of risk associated with perturbations in body fat distribution. A.C.F. reports grant support from Abbott Laboratories and is co-founder of Goodpath. The remaining authors declare no competing interests.
