## [Peer Review File · Nature Communications]

Clinical utility of polygenic scores for cardiometabolic disease in ArabsREVIEWER COMMENTS

Reviewer #1 (Remarks to the Author):

The authors studied the performance of polygenic scores (PGS/PRS) for cardiometabolic phenotypes (and height) in a study of an Arab population. They further propose a framework for optimizing PGS in a new population. This framework expresses what is often done in the field in practice, in a comprehensive and thorough manner and with some additional useful nuances. The manuscript is well written.

Comments, not ordered by any criterion:

- I think the authors should remove mentioned of novelty, as there are many papers reporting similar ideas, at least in part, including for using PCs across datasets.
- Propensity score matching of individuals between UKB and the Saudi dataset should be described.
- Which dataset/reference panel was used for LD by the various PRS software?
- Clinical utility analysis of polygenic scores: the interpretation of the percentiles as well as effect sizes is unclear, given that the sample is from a case-control study of CAD. The population is biased so these are not interpretable. Do the authors have any hypothesis regarding proportions of individuals with top quintile of risk? Also, is having 83.6% of individuals being at the top quintile of any score surprising? Or is it just a statistical inevitability?
- Description of high genomic risk of obesity, systolic BP hypertension, etc — it seems like the PGS are of BMI, SBP, etc, not of the binary measure. So the interpretation of having high genetic risk of the specific dichotomized phenotypes is not accurate.
- Study of interplay of PGS with conventional risk factors — it is not the right population to ask this question. Ideally this will be studied in a general population, rather than in a population enriched for CAD cases. While it is true that OR have similar estimates in case-control studies of the *primary outcome* and in population-based studies, this may not be the case when comparing an OR of a *secondary outcome* from a case-control study (i.e. not the primary outcome on which individuals were sample from) to that corresponding OR from a population-based study. Please see literature about secondary outcomes.
- It is a nice result that Arab-optimized PGS performed better in UK Arab-matched individuals compared to other PRS. However, a concern is that this results may be an artifact of the PGS selection, to some extent. For example, the European-derived polygenic scores may be already not optimal (e.g. based on older GWAS). Additional useful results and data to show:
 - 1) In table 2, provide information about the European-defined PGS (GWAS used to construct them/select variants and weights, software used to derive).
 - 2) Report how Arab-optimized matched PGS performed compared to the European-derived PGS in UKB Whites.
 - 3) Report the same comparisons in UKB matched-arabs.
- Please write "White individuals" and "Black individuals", rather than "blacks" and "whites".
- Discussion statement about continuous view of ancestry seems inappropriate, given that the authors have been dichotomizing ancestry — creating groups of Arabs, defining UKB individuals to the Arab

cluster, etc. It is true that this is not based on continental ancestry, and I also realize it is popular to talk about continuous ancestry, but here it not accurate.

- The use of PCs is inconsistent. Sometimes 10, 4, and 5. Given that PCs are derived using 1000 genome, at least 10 are needed to separate individuals of diverse genetic ancestries.

- What does the command/option `--thin-indiv-count` in PLINK do? It is great to report how things are done, but it is also important to explain what they mean/why they are done.

- Comparison to individuals of European ancestry in UKB: why use self-report and not ancestry inferred from 1000 genome?

Abstract:

- Writing that 83.6% of individuals having high genetic risk for something — is meaningless. Depends on the definition of risk, and some of the outcomes may not be diseases (BMI).

- “acted independently” — you mean had independent effect or additional association.

Reviewer #2 (Remarks to the Author):

I enjoyed reading the manuscript, “Clinical utility of polygenic scores for cardiometabolic disease in Arabs,” by Shim and colleagues. Here, the authors show that current polygenic scores for coronary artery disease, type 2 diabetes, and several quantitative traits perform less well in a Saudi Arabian cohort compared to their performance in Europeans. They go on to develop new scores from existing summary statistics, and these new scores show improved performance. This is an important contribution to growing efforts to improve equity in precision medicine. The authors frame their efforts as a pragmatic approach to polygenic score development. I think this framing is reasonable, but I would point out that much of their approach (e.g., steps 1-7 in Figure 1) are relatively standard in polygenic score research. The parts of their framework that I think stand out as novel and/or addressing open questions are steps 8-9 and steps 11-12. I applaud the authors for highlighting the often-overlooked fact that in order to use polygenic scores in the clinic, we need to define some reference to establish genetic risk percentiles and relative risk of the outcome. It remains uncertain how to best define such a reference. I also think the idea of using ancestry matching in biobanks to gather additional subjects for score validation is a good one and may be particularly helpful for populations that significantly under studied and underrepresented in genetics research.

Overall, I think this work is of clear interest, but I think there are some methodological areas that need clarification, justification, or improvement.

MAJOR

1. In the analysis of current polygenic score performance, the authors state, “Using one-to-one propensity matching for each trait, age, and sex to individuals of European ancestry from the UK Biobank, we generated an equally-sized cohort for each trait with similar characteristics and case-control ratio (Online Methods, Supplementary Table 2).” It is a little unclear what the authors mean by propensity matching. For each trait (e.g., CAD), did the authors calculate and match risks for each subject? Or are the traits themselves simply being matched? If propensity score matching was done, details of the model should be provided.

2. Is it possible that some of the polygenic scores used to compare performance in Europeans and Arabs were derived from GWAS that included subjects from the UK Biobank? If so, testing these scores in the UK Biobank may show inflated results due to overfitting. The authors should identify which scores may have sample overlap and note this as a limitation in the Discussion.

3. Figure 2c does not include the polygenic score for cardiomyopathy (PGS0020 51). Is there a reason to exclude this score from the comparison?

4. Some of the terminology used by the authors is a bit confusing. I admit, there is no universal standard for how to describe polygenic score development, but I think several leaders in the field have adopted relatively similar language (see for example Aragam & Natarajan *Circulation Research* 2020, Wand et al. *Nature* 2021, and O'Sullivan et al. *Circulation* 2022). What is most commonly described is using a "training" or "tuning" population for testing and choosing the best scores among several after varying some parameters. This best score is then applied to an independent "validation" population to demonstrate and estimate the performance. However, here the authors refer to their tuning population (eg Supp Table 6-7) as the "validation" population and they refer to the validation population (eg Supp Table 8) as the "test" population. Since there is no universal standard, if the authors have a strong reason to use this language, I leave it to their discretion. However, the current language may create confusion among readers.

5. The results shown in Table 2 may be misleading. The authors described a sound and well-accepted approach of tuning scores in one population and then validating ("testing") the best scores in an independent sample. The population being used in Table 2 is not clearly defined. I initially thought it represented an additional external validation cohort. However, it seems that this is simply the pooled set of subjects already used for tuning and validation. This is not common practice and is less rigorous than presenting results for an independent validation population, as is shown in Supp Table 8. Ideally, the cohort used to demonstrate and compare score performance should be independent of the tuning cohort used for selecting the best score. It would seem to me that the results shown in Supp Table 8 are the more rigorous result that should be presented, and for fair comparison, the European-derived scores should be tested in that subset.

6. The comment above about independence of the tuning (what the authors call "validation") and validation (what the authors call "testing") cohorts also applies to the subsequent analyses shown in Figure 3, Figure 4, etc.

7. The use of an external reference population to define percentages is a reasonable idea to put forward. What are the advantages/disadvantages of this approach? What would be the alternative? I think some would argue that in the ideal scenario the validation population should also serve as the reference. This approach works for an unselected prospective validation population. In this case, did the authors choose an outside reference because the validation cohort was selected for having cardiometabolic disease? Would the possibility of differences in population substructure affect the analysis? The authors could demonstrate the validity of their reference population through PC analysis or by showing the raw distribution of polygenic scores for the reference population compared to cases and controls in the target population. Presumably the reference population should have a distribution intermediary to cases and controls or more closely aligning to controls.

MINOR

1. The authors reference Wikipedia (ref 13) regarding the world Arab population. A more rigorous and stable reference would strengthen the manuscript.

2. The authors state in the results, "We first asked whether the most commonly used polygenic scores for cardiometabolic disease have reduced performance in Arabs (Figure 2 a,b)." However, Figure 2a

and 2b shows a map and PC analysis. Is this figure reference correct?

3. Supp Table 9 shows cohort characteristics that are related to or redundant to data shown in Table 1. It may make sense to simply have one table that is referenced throughout.

Response to Reviewers' Comments for NCOMMS-20-07162-T
"Clinical utility of polygenic scores for cardiometabolic disease in Arabs"

Reviewer #1 (Remarks to the Author):

The authors studied the performance of polygenic scores (PGS/PRS) for cardiometabolic phenotypes (and height) in a study of an Arab population. They further propose a framework for optimizing PGS in a new population. This framework expresses what is often done in the field in practice, in a comprehensive and thorough manner and with some additional useful nuances. The manuscript is well written.

Author Reply: We are grateful for the Reviewer's careful consideration of our manuscript. We have sought to address the useful comments as detailed below.

Comments, not ordered by any criterion:

- I think the authors should remove mention of novelty, as there are many papers reporting similar ideas, at least in part, including for using PCs across datasets.

Author Reply: We agree with the reviewer and removed the mention of novelty from the updated manuscript.

Manuscript Changes: We removed the word "novel" from the following sentence:

"Fourth, we propose a ~~novel~~ method for identifying ancestrally-matched individuals to the target sub-population in large biobanks .."

- Propensity score matching of individuals between UKB and the Saudi dataset should be described.

Author Reply: We agree that the manuscript would benefit from additional description of the matched cohort of the UK biobank in the "Results" and "Online Methods" sections.

Manuscript Changes: In response, we have modified the following sentence in the Results section from:

"Using one-to-one propensity matching for each trait, age, and sex to individuals of European ancestry from the UK Biobank, we generated an equally-sized cohort for each trait with similar characteristics and case-control ratio"

to

"For each of the 10 traits of interest, we used one-to-one nearest neighbor matching to generate an equal-sized cohort of individuals of European ancestry from the UK Biobank with

comparable age, sex, and case-control ratio for categorical matching traits or mean value for continuous matching traits (Online Methods, Supplementary Table 2)."

We also added the following text to the Online Methods:

"For each of the 10 traits, we derived a matched cohort of equal size from individuals of European ancestry in the UK Biobank using the R package "MatchIt".⁵⁶ Briefly, this technique uses 1:1 nearest neighbor matching to generate an equal sized cohort that is as similar to the study population as possible in terms of age, sex, and case-control ratio for categorical traits, or mean value for continuous traits."

56. Ho, D., Imai, K., King, G., & Stuart, E. A. MatchIt: Nonparametric Preprocessing for Parametric Causal Inference. *Journal of Statistical Software*, 42(8), 1–28 (2011).

- Which dataset/reference panel was used for LD by the various PRS software?

Author Reply: We agree that providing details on the LD reference panel used for each software is important. We have now included LD reference information for each approach in the 'LD reference' column of Supplementary Table 4 and pointed to that in the manuscript.

Manuscript Changes: We added the following sentence in the 'Arab-optimized polygenic scores on par with European scores' section of the Results:

"Each method used in this study was briefly described along with its parameters and LD reference panel (Supplementary Table 4)."

We also added the following text to the 'Polygenic score calculation' section of the Online Methods:

"We utilized an in-sample LD matrix for PRSice-2, and external LD reference panels constructed using the 1000 Genomes Project Phase 3 dataset for the remaining approaches. For LDpred2, lassosum2, and PRS-CS, we used an LD reference panel built using European samples (N=503). For PRS-CSx, we used LD reference panels created from the Arab population of this study (N=6,416) and five super populations in the 1000 Genomes Project."

- Clinical utility analysis of polygenic scores: the interpretation of the percentiles as well as effect sizes is unclear, given that the sample is from a case-control study of CAD. The population is biased so these are not interpretable. Do the authors have any hypothesis regarding proportions of individuals with top quintile of risk? Also, is having 83.6% of individuals being at the top quintile of any score surprising? Or is it just a statistical inevitability?

Author Reply: The reviewer raises three important points in this comment:

First, can effect size from case-control study be used? From a pragmatic standpoint, the case-control design in this study allows an estimate of effect size for multiple traits using a relatively smaller sample size. This approach has been used in previous studies, and in general estimates of effect size from case-control studies or hospital biobank studies (enriched for cases) vs. population cohorts are comparable (PMID 32498804 and 33284643). However, it is important to acknowledge this in the limitations for the current study.

*“The study has few limitations that are important to note. First, **this is cross-sectional case-control study without longitudinal follow-up, which limits evaluation of risk longitudinally and population estimates of clinico-genomic risk.**”*

Second, how should percentiles be defined? Using the same case-control population used to derive the effect size to generate percentiles is likely to result in a skewed population distribution. Instead, we opted to use a population reference with no known disease and that is representative of the general population. Prior efforts have used this approach such as in the South Asian population.

Wang M et al. Validation of a Genome-Wide Polygenic Score for Coronary Artery Disease in South Asians. *J Am Coll Cardiol.* 2020 Aug 11;76(6):703-714. (PMID 32762905)

Third, we agree with the reviewer’s comment regarding potential statistical inevitability of using the top quintile to define the proportion of individuals with high genomic risk in this case. This estimate does not reflect the population reference and could be misleading for readers. In the revised manuscript we have removed this section of the results.

Manuscript Changes: We removed the following section of the results from the abstract and manuscript text.

“Polygenic scores ~~identified 83.6% of individuals as having high genomic risk (top quintile of population distribution) for at least one disease, were associated with disease independent of acted independently from conventional risk factors, and also associated with disease severity.~~”

“The top quintile of the polygenic score is a commonly used threshold to identify 20% of the population with high genomic risk for disease. In this Arab cohort with cardiovascular disease, 83.6% (4,016 of 4,805) participants carried a high genomic risk for at least one of 7 traits: 29.8% (1,039 of 3,491) individuals had high genomic risk for CAD, 34.0% (1,014 of 2,979) had high genomic risk for type 2 diabetes, and 29.4% (141 of 479) had high genomic risk for cardiomyopathy. Also, 25.9% (551 of 2,126) individuals had high genomic risk of obesity (defined as BMI \geq 30 kg/m²), 25.5% (455 of 1,783) had high genomic risk for hypercholesterolemia (defined as estimated untreated LDL cholesterol \geq 140 mg/dL), 24.0% (932 of 3,879) had high genomic risk for systolic hypertension (defined as estimated untreated systolic blood pressure \geq 130 mmHg), and 23.9% (741 of 3,096) had high genomic risk for diastolic hypertension (defined as estimated untreated diastolic blood pressure \geq 80 mmHg).”

- Description of high genomic risk of obesity, systolic BP hypertension, etc — it seems like the PGS are of BMI, SBP, etc, not of the binary measure. So the interpretation of having high genetic risk of the specific dichotomized phenotypes is not accurate.

Author Reply: We agree with the reviewer and as stated above, we have removed the reference of the continuous trait polygenic score to define a proportion of patients with a binary disease generated based on this continuous trait.

Manuscript Changes: We have removed the following paragraph from the results:

“The top quintile of the polygenic score is a commonly used threshold to identify 20% of the population with high genomic risk for disease. In this Arab cohort with cardiovascular disease, 83.6% (4,016 of 4,805) participants carried a high genomic risk for at least one of 7 traits: 29.8% (1,039 of 3,491) individuals had high genomic risk for CAD, 34.0% (1,014 of 2,979) had high genomic risk for type 2 diabetes, and 29.4% (141 of 479) had high genomic risk for cardiomyopathy. Also, 25.9% (551 of 2,126) individuals had high genomic risk of obesity (defined as BMI ≥ 30 kg/m²), 25.5% (455 of 1,783) had high genomic risk for hypercholesterolemia (defined as estimated untreated LDL cholesterol ≥ 140 mg/dL), 24.0% (932 of 3,879) had high genomic risk for systolic hypertension (defined as estimated untreated systolic blood pressure ≥ 130 mmHg), and 23.9% (741 of 3,096) had high genomic risk for diastolic hypertension (defined as estimated untreated diastolic blood pressure ≥ 80 mmHg).”

- Study of interplay of PGS with conventional risk factors — it is not the right population to ask this question. Ideally this will be studied in a general population, rather than in a population enriched for CAD cases. While it is true that OR have similar estimates in case-control studies of the *primary outcome* and in population-based studies, this may not be the case when comparing an OR of a *secondary outcome* from a case-control study (i.e. not the primary outcome on which individuals were sampled from) to that corresponding OR from a population-based study. Please see literature about secondary outcomes.

Author Reply: The study cohort consists of individuals referred to a cardiologist for work-up of coronary artery disease and is as such enriched for diabetes. Our primary analysis of conventional risk factors with polygenic score is for CAD with an additional analysis for diabetes. Specifically, we evaluate for 1) whether the effect of the polygenic score on the CAD or diabetes remains after adjusting for conventional risk factors, 2) whether there is a statistical interaction between the score and conventional risk factors, and 3) whether the effect of the polygenic score is consistent among known clinical subgroups. It is precisely because of what the reviewer pointed out that we avoid making any claims about clinico-genomic risk in this cross-sectional study.

Manuscript Changes: We highlight the following in the limitations of the manuscript:

*“The study has few limitations that are important to note. First, this is cross-sectional case-control study without longitudinal follow-up, which **limits evaluation of risk longitudinally and population estimates of clinico-genomic risk.**”*

- It is a nice result that Arab-optimized PGS performed better in UK Arab-matched individuals compared to other PRS. However, a concern is that these results may be an artifact of the PGS selection, to some extent. For example, the European-derived polygenic scores may be already not optimal (e.g. based on older GWAS). Additional useful results and data to show:

- 1) In table 2, provide information about the European-defined PGS (GWAS used to construct them/select variants and weights, software used to derive).
- 2) Report how Arab-optimized matched PGS performed compared to the European-derived PGS in UKB Whites.
- 3) Report the same comparisons in UKB matched-arabs.

Author Reply: We appreciate and agree with the concern. In our initial submission, we had attempted to use the most recent and largest samples size PGS from the catalog (listed in supplementary table 1), but we have now calculated most scores in the PGS catalog for each continuous trait (total 36 scores), and observe variability in performance, mostly driven by scores using GWAS data from the UK Biobank having an inflated effect size in the UK Biobank.

In the figure below, we show the performance of Arab-optimized scores (in red) vs. multiple scores from the PGS Catalog (green) for the UK Biobank Arab-matched cohort (N=420, diamond) vs. 1000 randomly selected groups of European ancestry from the UK Biobank (boxplot). The scores in light blue were included in the initial manuscript submission. The scores in dark blue are the European-derived scores used in the revised manuscript, which were updated with scores developed from GWAS that excluded the UK biobank.

In the revised manuscript, we select the European-derived score for comparison as ones that have specifically not been derived in the UK Biobank and at the same time representing the most recent and largest GWAS. The table below shows the list of European-derived scores used in the initial submission vs. the updated ones in this revision.

Trait	Initial submission		Revised manuscript	
	PGS Catalog ID	GWAS source	PGS Catalog ID	GWAS source
Coronary artery disease	PGS000013	CARDIoGRAMplus C4D	Same	
Type 2 diabetes	PGS000014	DIAGRAM, EPIC, GERA	Same	
Cardiomyopathy	PGS002051	UKB	Same	
LDL Cholesterol	PGS000888	GLGC (Multi-ancestry)	PGS000892	GLGC (EUR)
HDL Cholesterol	PGS002329	UKB	PGS002781	GLGC (Multi-ancestry)
Triglyceride	PGS002353	UKB	PGS002784	GLGC (Multi-ancestry)
Systolic blood pressure	PGS002349	UKB	PGS002238	MVP, BBJ, UKB
Diastolic blood pressure	PGS002322	UKB	PGS002239	MVP, BBJ, UKB
Body mass index	PGS000027	GIANT	Same	
Height	PGS002332	UKB	PGS002804	GIANT

The main findings of the manuscript continue to hold with the updated scores. Mainly, there is a decrease in performance in those published scores among Arabs in this study and in Arab-matched participants in the UK Biobank, compared to individuals of European ancestry in the UK Biobank. The Arab-optimized scores in this study also improve on this performance in the Arab cohort. However, improvement in performance using the Arab-optimized scores in the 420 Arab-matched in UK Biobank is not observed consistently for all traits. Specifically, it is observed for triglycerides, systolic blood pressure, and diastolic blood pressure. It is not observed for body mass index, height, LDL cholesterol, and HDL cholesterol.

Manuscript Changes: We have now added the derivation method of European-derived scores to Table 2. In Supplementary Table S1, we include details about the GWAS source of the score being used and call for this supplementary table in the legend of Table 2.

The updated Table 2 is included below:

Traits	European-derived polygenic scores					Arab optimized polygenic scores			
	PGS Catalog ID	Derivation Method	No. of variants in PRS	OR per SD (95% CI)	AUC (95% CI)	Derivation Method	No. of variants in PRS	OR per SD (95% CI)	AUC (95% CI)
Coronary artery disease	PGS000013	LDpred	5,706,928	1.41 (1.31-1.50)	0.7909 (0.7726-0.8091)	lassosum2	10,440	1.51 (1.42-1.61)	0.7950 (0.7768-0.8132)
Type 2 diabetes	PGS000014	LDpred	5,786,938	1.41 (1.32-1.49)	0.7054 (0.6854-0.7255)	PRS-CS	1,068,166	1.83 (1.74-1.92)	0.7384 (0.7194-0.7574)
Cardiomyopathy	PGS002051	LDpred2	621,802	1.01 (0.88-1.16)	0.6277 (0.5890-0.6665)	LDpred2	1,010,014	1.34 (1.13-1.64)	0.6453 (0.6086-0.6819)
Continuous traits	PGS Catalog ID	Derivation Method	No. of variants in PRS	Effect size (SE)	Adjusted R-sq	Derivation Method	No. of variants in PRS	Effect size (SE)	Adjusted R-sq
LDL Cholesterol [mg/dL]	PGS000892	PRS-CS	1,068,974	10.06 (1.10)	0.0405	PRS-CSx	1,069,677	9.40 (1.10)	0.0358
HDL Cholesterol [mg/dL]	PGS002781	PRS-CS	1,112,500	3.49 (0.27)	0.1351	PRS-CSx	1,069,677	3.67 (0.27)	0.1424
Triglyceride [mg/dL]	PGS002784	Pruning and Thresholding	12,709	25.19 (2.34)	0.0682	lassosum2	54,623	29.32 (2.31)	0.0857
Systolic blood pressure [mmHg]	PGS002238	PRS-CS	1,069,203	1.08 (0.45)	0.0961	PRS-CS	1,056,790	3.10 (0.44)	0.1108
Diastolic blood pressure [mmHg]	PGS002239	PRS-CS	1,069,056	0.71 (0.26)	0.0249	lassosum2	25,857	1.80 (0.26)	0.0397
Body mass index [kg/m ²]	PGS000027	LDpred	2,015,065	0.97 (0.11)	0.0790	PRS-CSx	1,067,771	1.18 (0.11)	0.0919
Height [m]	PGS002804	SBayesC	1,054,056	0.027 (0.0013)	0.5299	PRS-CSx	1,067,771	0.026 (0.0013)	0.5202

Performance of European-derived polygenic scores and Arab-optimized polygenic scores for each trait are shown in the validation dataset of Arabs (N=2,699). European-derived polygenic scores were calculated using scoring files from The Polygenic Score (PGS) Catalog with datasets consisting primarily of European ancestry **and not derived from the UK Biobank. Further information about the scoring files, including their GWAS sources, is included in Supplementary Table 1.** Arab-optimized polygenic scores were derived in this study. OR per SD and AUC were determined using a logistic regression model adjusted for age, sex, array version, and the first 10 principal components of ancestry. Effect size and adjusted R^2 values for continuous traits were determined using a linear regression model with similar covariates. (OR per SD: odd ratio per standard deviation, AUC: area under the receiver operating characteristic curve, CI: confidence interval, SE: standard error).

We also added the following sentence to the Online Methods:

“We prioritized scores that are not derived from the UK Biobank to enable comparison of performance of those scores in Arabs to individuals of European ancestry from the UK Biobank.”

We also edited the results text discussing the performance of Arab-optimized polygenic scores in UK Arab-matched individuals compared to European-derived scores as follows:

“We tested the performance of both European-derived scores and our Arab-optimized scores for continuous traits in this “Arab-matched” cohort in the UK Biobank compared to their performance in 1000 randomly selected groups of European ancestry from the UK Biobank. European-derived scores had ~~2741%~~ reduced performance in the Arab-matched cohort compared to the European-ancestry cohort. ~~The~~However, the optimized scores from our study improved performance among both Arab-matched and European ancestry participants in the UK Biobank for systolic blood pressure, diastolic blood pressure and triglycerides, but did not result in a significant improvement in performance for the other traits. ~~— to a 31% residual decrement across all 7 traits. Compared with European-derived scores, the Arab-optimized scores increased the association performance in the Arab-matched cohort relative to that in the European-ancestry cohort substantially across all seven traits~~ (Supplementary Figure 75). Heterogeneity of effect due to polygenic score selection used for comparison and small sample size of the Arab-matched cohort is a limitation of this analysis.”

- Please write “White individuals” and “Black individuals”, rather than “blacks” and “whites”.

Author Reply: We agree with the reviewer and changed the words in the updated manuscript.

Manuscript Changes: We have revised the following sentence from the Discussion:

*“This has been shown among ~~blacks~~ **Black individuals** in the USA in a recent study whereby the polygenic score improved the estimation of absolute risk of myocardial infarction in ~~blacks~~ **Black individuals** more than in ~~whites~~ **White individuals** despite a weaker association of the score with disease.”*

- Discussion statement about continuous view of ancestry seems inappropriate, given that the authors have been dichotomizing ancestry — creating groups of Arabs, defining UKB individuals to the Arab cluster, etc. It is true that this is not based on continental ancestry, and I also realize it is popular to talk about continuous ancestry, but here it not accurate.

Author Reply: We agree with the reviewer and have removed the statement from the discussion.

Manuscript Changes: We removed the following statement:

“Genetic ancestry in the case of polygenic scores might be a more appropriate measure to advance cross-ancestry implementation, ~~and recent efforts are helping view genetic ancestry as a continuous measure rather than continental subgroups.~~”

- The use of PCs is inconsistent. Sometimes 10, 4, and 5. Given that PCs are derived using 1000 genome, at least 10 are needed to separate individuals of diverse genetic ancestries.

Author Reply: We agree with the reviewer. In this revision, we have updated the number of PCs utilized in all the analyses to the first 10 PCs.

Manuscript Changes: We have made the appropriate modifications to all tables and text. Specifically, the following Table 2, Figure 2 (c and d), Figure 3, Figure 4, Supplementary Table 2, Supplementary Table 6, Supplementary Table 7, Supplementary Table 8, Supplementary Table 9, and Supplementary Table 10 were updated. Overall, this did not significantly change any of the findings of the study.

We also updated the text to reflect that 10 PCs were used in the analyses:

*“Identifying best-performing polygenic score
Maximum area-under-curve (AUC) or adjusted R-squared of the score evaluated in a regression model with age, sex, array version, and **first 10 ~~four~~ principal components** .. “*

*“Defining polygenic score percentiles
In a linear regression model, the ancestry-adjusted polygenic score was the raw polygenic score adjusted by the **first 10 ~~five~~ principal components of ancestry.**”*

- What does the command/option —thin-indiv-count in PLINK do? It is great to report how things are done, but it is also important to explain what they mean/why they are done.

Author Reply: We appreciate the reviewer's comment and agree that additional clarification is needed. This *--thin-indiv-count <n>* in PLINK 2.0 randomly removes samples from a dataset until n samples remain.

Manuscript Changes: In response, we have modified the following sentence in the Online Methods as follows:

"To determine the optimal polygenic score model, the study population was split into training and validation datasets. Using --thin-indiv-count <n> in PLINK 2.0, which randomly removes samples until n samples remain, we defined n=2,700. This resulted in a training dataset of 2,700 samples. The remaining 2,699 samples were utilized as a validation dataset (Supplementary Table 5)."

- Comparison to individuals of European ancestry in UKB: why use self-report and not ancestry inferred from 1000 genome?

Author Reply: We agree with the reviewer that genetically inferred ancestry in the UK Biobank should be utilized and have updated the analysis accordingly.

Manuscript Changes: We have now selected individuals of European ancestry from the UK Biobank using genetically inferred ancestry rather than the self-report.

The supplementary figure 7 has been updated with this analysis, with no significant change to the overall findings.

The text of the manuscript was also updated as follows:

*"To compare performance of scores in Arab-matched samples to individuals of European ancestry, we defined a European ancestry cohort **using a genetically defined Caucasian ethnic group reported in the UK Biobank (field 2206) self-report of 'British', 'Irish', or 'Any other white background'** (N=223,901).*

Abstract:

- Writing that 83.6% of individuals having high genetic risk for something — is meaningless. Depends on the definition of risk, and some of the outcomes may not be diseases (BMI).

- "acted independently" — you mean had independent effect or additional association.

Author Reply: We agree and as stated previously have removed this part of the results from the manuscript.

Manuscript Changes: We changed the sentence in the abstract as follows:

“Polygenic scores ~~identified 83.6% of individuals as having high genomic risk (top quintile of population distribution) for at least one disease, were associated with disease independent of acted independently from~~ conventional risk factors, and also associated with disease severity.”

Reviewer #2 (Remarks to the Author):

I enjoyed reading the manuscript, “Clinical utility of polygenic scores for cardiometabolic disease in Arabs,” by Shim and colleagues. Here, the authors show that current polygenic scores for coronary artery disease, type 2 diabetes, and several quantitative traits perform less well in a Saudi Arabian cohort compared to their performance in Europeans. They go on to develop new scores from existing summary statistics, and these new scores show improved performance. This is an important contribution to growing efforts to improve equity in precision medicine. The authors frame their efforts as a pragmatic approach to polygenic score development. I think this framing is reasonable, but I would point out that much of their approach (e.g., steps 1-7 in Figure 1) are relatively standard in polygenic score research. The parts of their framework that I think stand out as novel and/or addressing open questions are steps 8-9 and steps 11-12. I applaud the authors for highlighting the often-overlooked fact that in order to use polygenic scores in the clinic, we need to define some reference to establish genetic risk percentiles and relative risk of the outcome. It remains uncertain how to best define such a reference. I also think the idea of using ancestry matching in biobanks to gather additional subjects for score validation is a good one and may be particularly helpful for populations that significantly under studied and underrepresented in genetics research.

Overall, I think this work is of clear interest, but I think there are some methodological areas that need clarification, justification, or improvement.

Author Reply: We appreciate the Reviewer’s thoughtful evaluation of our manuscript.

MAJOR

1. In the analysis of current polygenic score performance, the authors state, “Using one-to-one propensity matching for each trait, age, and sex to individuals of European ancestry from the UK Biobank, we generated an equally-sized cohort for each trait with similar characteristics and case-control ratio (Online Methods, Supplementary Table 2).” It is a little unclear what the authors mean by propensity matching. For each trait (e.g., CAD), did the authors calculate and match risks for each subject? Or are the traits themselves simply being matched? If propensity score matching was done, details of the model should be provided.

Author Reply: We agree that the manuscript would benefit from additional description of the matched cohort of the UK biobank in the “Results” and “Online Methods” sections.

Manuscript Changes: In response, we have modified the following sentence in the Results section from:

~~“Using one-to-one propensity matching for each trait, age, and sex to individuals of European ancestry from the UK Biobank, we generated an equally-sized cohort for each trait with similar characteristics and case-control ratio”~~

to

“For each of the 10 traits of interest, we used one-to-one nearest neighbor matching to generate an equal-sized cohort of individuals of European ancestry from the UK Biobank with comparable age, sex, and case-control ratio for categorical matching traits or mean value for continuous matching traits (Online Methods, Supplementary Table 2).”

We also added the following text to the Online Methods:

“For each of the 10 traits, we derived a matched cohort of equal size from individuals of European ancestry in the UK Biobank using the R package "MatchIt".⁵⁶ Briefly, this technique uses 1:1 nearest neighbor matching to generate an equal sized cohort that is as similar to the study population as possible in terms of age, sex, and case-control ratio for categorical traits, or mean value for continuous traits.”

56. Ho, D., Imai, K., King, G., & Stuart, E. A. MatchIt: Nonparametric Preprocessing for Parametric Causal Inference. *Journal of Statistical Software*, 42(8), 1–28 (2011).

2. Is it possible that some of the polygenic scores used to compare performance in Europeans and Arabs were derived from GWAS that included subjects from the UK Biobank? If so, testing these scores in the UK Biobank may show inflated results due to overfitting. The authors should identify which scores may have sample overlap and note this as a limitation in the Discussion.

Author Reply: We appreciate and agree with the concern. We calculated most scores in the PGS catalog for each trait (total 36 scores), including multiple new scores that have been deposited since our initial submission, and observe variability in performance, mostly driven by scores using GWAS data from the UK Biobank having an inflated effect size in the UK Biobank.

In the figure below, we show the performance of Arab-optimized scores (in red) vs. multiple scores from the PGS Catalog (green) for the UK Biobank Arab-matched cohort (N=420, diamond) vs. 1000 randomly selected groups of European ancestry from the UK Biobank (boxplot). The scores in light blue were included in the initial manuscript submission. The scores in dark blue are the European-derived scores used in the revised manuscript, which were updated with scores developed from GWAS that excluded the UK biobank.

In the revised manuscript, we select the European-derived score for comparison as ones that have specifically not been derived in the UK Biobank and at the same time representing the most recent and largest GWAS. The table below shows the list of European-derived scores used in the initial submission vs. the updates ones in this revision.

Trait	Initial submission		Revised manuscript	
	PGS Catalog ID	GWAS source	PGS Catalog ID	GWAS source
Coronary artery disease	PGS000013	CARDIoGRAMplus C4D	Same	
Type 2 diabetes	PGS000014	DIAGRAM, EPIC, GERA	Same	
Cardiomyopathy	PGS002051	UKB	Same	
LDL Cholesterol	PGS000888	GLGC (Multi-ancestry)	PGS000892	GLGC (EUR)
HDL Cholesterol	PGS002329	UKB	PGS002781	GLGC (Multi-ancestry)
Triglyceride	PGS002353	UKB	PGS002784	GLGC (Multi-ancestry)
Systolic blood pressure	PGS002349	UKB	PGS002238	MVP, BBJ, UKB
Diastolic blood pressure	PGS002322	UKB	PGS002239	MVP, BBJ, UKB
Body mass index	PGS000027	GIANT	Same	
Height	PGS002332	UKB	PGS002804	GIANT

Manuscript Changes: We have updated the European-derived polygenic scores used in the primary analyses for LDL cholesterol, HDL cholesterol, triglycerides, systolic blood pressure, diastolic blood pressure, and height to ensure that all scores are not derived using UK Biobank data. We included details about the GWAS source of the score being used in Supplementary Table S1 and added the information in the legend of Table 2. Also, the tables, figures, and text have been modified to reflect the updated scoring file results. The overall findings remain the same after updating those scores.

Supplementary Table 1. List of scoring files for polygenic scores from the PGS catalog

Trait	PGS catalog ID	Citation	Derivation Strategy	Tuning Parameter	No. variants available	Source of variant associations (GWAS)
Coronary artery disease	PGS000013	Khera AV et al. Nat Genet (2018)	LDpred	$\rho = 0.001$	6,630,150	CARDIoGRAM plusC4D
Type 2 diabetes	PGS000014	Khera AV et al. Nat Genet (2018)	LDpred	$\rho = 0.01$	6,917,436	DIAGRAM, EPIC, GERA

Cardiomyopathy	PGS0020 51	Privé F et al. Am J Hum Genet (2022)	LDpred2	auto	642,241	UKB
LDL cholesterol	PGS0008 92	Graham SE et al. Nature (2021)	PRS-CS	auto	1,239,184	GLGC
HDL cholesterol	PGS0027 81	Kanoni S et al. Genome Biol (2021)	PRS-CS	auto	1,239,184	GLGC
Triglycerides	PGS0027 84	Kanoni S et al. Genome Biol (2021)	Pruning and Thresholding	$r^2=0.1$, p -value= $5e-3$, 500kb	30,071	GLGC
Systolic blood pressure	PGS0022 38	Breyear JH et al. Circ Genom Precis Med (2022)	PRS-CS after P-value thresholding	p -value= $1e-1$	1,119,444	MVP, UKB, BBJ
Diastolic blood pressure	PGS0022 39	Breyear JH et al. Circ Genom Precis Med (2022)	PRS-CS	p -value= $1e-1$	1,119,054	MVP, UKB, BBJ
Body mass index	PGS0000 27	Khera AV et al. Cell (2019)	LDpred	$\rho = 0.03$	2,100,302	GIANT
Height	PGS0028 04	Yengo L et al. Nature (2022)	SBayesC	-	1,103,042	GIANT

3. Figure 2c does not include the polygenic score for cardiomyopathy (PGS002051). Is there a reason to exclude this score from the comparison?

Author Reply: We appreciate the reviewer’s comment. The only European-derived score for cardiomyopathy (PGS002051) was derived using UK Biobank GWAS, and as such its effect size estimate in UK Biobank is inflated (Odds ratio per standard deviation of 3.13 (95% CI 2.98-3.27)). A comparison would not be accurate.

Manuscript Changes: We added a note to the legend of figure 2c to clarify:

“The polygenic score for cardiomyopathy (PGS002051) has been derived in the UK Biobank where it has an inflated estimate of effect size making a comparison to Arabs inaccurate.”

4. Some of the terminology used by the authors is a bit confusing. I admit, there is no universal standard for how to describe polygenic score development, but I think several leaders in the field have adopted relatively similar language (see for example Aragam & Natarajan Circulation Research 2020, Wand et al. Nature 2021, and O’Sullivan et al. Circulation 2022). What is most commonly described is using a “training” or “tuning” population for testing and choosing the best scores among several after varying some parameters. This best score is then applied to an independent “validation” population to demonstrate and estimate the performance. However, here the authors refer to their tuning population (eg Supp Table 6-7) as the “validation” population and they refer to the validation population (eg Supp Table 8) as the “test” population. Since there is no universal standard, if the authors have a strong

reason to use this language, I leave it to their discretion. However, the current language may create confusion among readers.

Author Reply: We appreciate the thoughtful comment by the reviewer and agree that using the language “training” and “validation” is most consistent with prior reports by leaders in the field. We have updated the manuscript using this language to be consistent with the literature and avoid any confusion in the field.

Manuscript Changes: We updated Figure 1 to use “training” and “validation”

We also updated the relevant sections of the text. Representative examples are shown below:

*“Second, the target population is split into **training validation** and **validation testing** sets. The performance of the different scores for a trait is compared in the **training validation** set, and the best-performing score is selected and reported in the **validation testing** set where its association with the trait of interest is reported using regression models.”*

*“The best-performing score was then tested in a **validation testing** set (N=2,699)”*

*“Maximum area-under-curve (AUC) or adjusted R-squared of the score evaluated in a regression model with age, sex, array version, and first 10 principal components in a **training validation** set was used to determine the best model for each trait”*

5. The results shown in Table 2 may be misleading. The authors described a sound and well-accepted approach of tuning scores in one population and then validating (“testing”) the best scores in an independent sample. The population being used in Table 2 is not clearly defined. I initially thought it represented an additional external validation cohort. However, it seems that this is simply the pooled

set of subjects already used for tuning and validation. This is not common practice and is less rigorous than presenting results for an independent validation population, as is shown in Supp Table 8. Ideally, the cohort used to demonstrate and compare score performance should be independent of the tuning cohort used for selecting the best score. It would seem to me that the results shown in Supp Table 8 are the more rigorous result that should be presented, and for fair comparison, the European-derived scores should be tested in that subset.

Author Reply: We agree with the reviewer’s comment and have updated the main results of the manuscript to show those in the validation set (N=2,699) only.

Manuscript Changes: We updated Table 2 whereby the main results are now shown for the validation set (N=2,699). We also updated the numbers in the manuscript text.

Traits	European-derived polygenic scores					Arab optimized polygenic scores			
	PGS Catalog ID	Derivation Method	No. of variants in PRS	OR per SD (95% CI)	AUC (95% CI)	Derivation Method	No. of variants in PRS	OR per SD (95% CI)	AUC (95% CI)
Coronary artery disease	PGS000013	LDpred	5,706,928	1.41 (1.31-1.50)	0.7909 (0.7726-0.8091)	lassosum ₂	10,440	1.51 (1.42-1.61)	0.7950 (0.7768-0.8132)
Type 2 diabetes	PGS000014	LDpred	5,786,938	1.41 (1.32-1.49)	0.7054 (0.6854-0.7255)	PRS-CS	1,068,166	1.83 (1.74-1.92)	0.7384 (0.7194-0.7574)
Cardiomyopathy	PGS002051	LDpred2	621,802	1.01 (0.88-1.16)	0.6277 (0.5890-0.6665)	LDpred2	1,010,014	1.34 (1.13-1.64)	0.6453 (0.6086-0.6819)
Continuous traits	PGS Catalog ID	Derivation Method	No. of variants in PRS	Effect size (SE)	Adjusted R-sq	Derivation Method	No. of variants in PRS	Effect size (SE)	Adjusted R-sq
LDL Cholesterol [mg/dL]	PGS000892	PRS-CS	1,068,974	10.06 (1.10)	0.0405	PRS-CSx	1,069,677	9.40 (1.10)	0.0358
HDL Cholesterol [mg/dL]	PGS002781	PRS-CS	1,112,500	3.49 (0.27)	0.1351	PRS-CSx	1,069,677	3.67 (0.27)	0.1424
Triglyceride [mg/dL]	PGS002784	Pruning and Thresholding	12,709	25.19 (2.34)	0.0682	lassosum ₂	54,623	29.32 (2.31)	0.0857
Systolic blood pressure [mmHg]	PGS002238	PRS-CS	1,069,203	1.08 (0.45)	0.0961	PRS-CS	1,056,790	3.10 (0.44)	0.1108
Diastolic blood pressure [mmHg]	PGS002239	PRS-CS	1,069,056	0.71 (0.26)	0.0249	lassosum ₂	25,857	1.80 (0.26)	0.0397

Body mass index [kg/m²]	PGS000027	LDpred	2,015,065	0.97 (0.11)	0.0790	PRS-CSx	1,067,771	1.18 (0.11)	0.0919
Height [m]	PGS002804	SBayesC	1,054,056	0.027 (0.0013)	0.5299	PRS-CSx	1,067,771	0.026 (0.0013)	0.5202

We also added the following description to the legend of Table 2:

“Performance of European-derived polygenic scores and Arab-optimized polygenic scores for each trait are shown in the validation dataset of Arabs (N=2,699)”

6. The comment above about independence of the tuning (what the authors call “validation”) and validation (what the authors call “testing”) cohorts also applies to the subsequent analyses shown in Figure 3, Figure 4, etc.

Author Reply: We updated all main analyses in the manuscript to include only the validation set (N=2,699). Overall, the main results of the manuscript were not significantly different.

Manuscript Changes: We updated figure 3, figure 4, supplementary figure 4, supplementary figure 5, and supplementary table 8 to reflect data from the validation set only. We also updated the relevant text in the manuscript.

Figure 3. Stratification of cardiometabolic disease by polygenic scores among Arabs

Figure 4. Performance of coronary artery disease polygenic score in different patient subgroups

		N Total	N Cases	OR (95% CI)	P-value
Any coronary artery disease (CAD)		2,682	1,712	1.51 (1.42-1.61)	7.21×10^{-18}
Obstructive CAD		2,657	1,269	1.62 (1.53-1.71)	6.72×10^{-26}
Severity of obstructive CAD	One-vessel	-	445	1.41 (1.29-1.53)	6.20×10^{-9}
	Two-vessel	-	250	1.68 (1.53-1.83)	2.21×10^{-11}
	Three-vessel	-	574	1.78 (1.66-1.90)	2.62×10^{-21}
Age at CAD	> 55 years	1,372	1,072	1.44 (1.30-1.58)	2.04×10^{-7}
	≤ 55 years	1,310	640	1.58 (1.45-1.72)	3.37×10^{-11}
Sex	Female	953	442	1.46 (1.31-1.61)	7.55×10^{-7}
	Male	1,729	1,270	1.55 (1.43-1.67)	1.97×10^{-12}
Type 2 diabetes	Nondiabetic	1,204	584	1.40 (1.26-1.54)	2.44×10^{-6}
	Diabetic	1,465	1,127	1.57 (1.44-1.70)	4.00×10^{-11}
Obesity	Nonobese	1,512	949	1.46 (1.33-1.59)	1.09×10^{-8}
	Obese	1,029	661	1.57 (1.42-1.72)	2.38×10^{-9}

1.00 1.25 1.50 1.75
Odds ratio

Supplementary Figure 4. Stratification of disease risk or measured trait by polygenic score

Supplementary Figure 5. Performance of type 2 diabetes polygenic score in different patient subgroups

Supplementary Table 8. The interplay of coronary artery disease polygenic score with conventional risk factors

Model	OR per SD	Lower 95% CI	Upper 95% CI	P-value	P-value for interaction
CAD ~ PRS + age + sex + batch + 10 PCs (baseline)	1.51	1.42	1.61	7.21E-18	
CAD ~ PRS + age + sex + batch + 10 PCs + Smoking + PRS*Smoking	1.50	1.38	1.62	3.82E-11	0.95
CAD ~ PRS + age + sex + batch + 10 PCs + Obesity + PRS*Obesity	1.46	1.33	1.59	8.69E-09	0.43
CAD ~ PRS + age + sex + batch + 10 PCs + SBP + PRS*SBP	1.33	0.72	1.94	3.60E-01	0.65
CAD ~ PRS + age + sex + batch + 10 PCs + DM + PRS*DM	1.39	1.25	1.52	1.97E-06	0.12
CAD ~ PRS + age + sex + batch + 10 PCs + LDL + PRS*LDL	1.70	1.41	1.99	3.62E-04	0.38

7. The use of an external reference population to define percentages is a reasonable idea to put forward. What are the advantages/disadvantages of this approach? What would be the alternative? I think some would argue that in the ideal scenario the validation population should also serve as the reference. This approach works for an unselected prospective validation population. In this case, did the authors choose an outside reference because the validation cohort was selected for having cardiometabolic disease? Would the possibility of differences in population substructure effect the analysis? The authors could demonstrate the validity of their reference population through PC analysis

or by showing the raw distribution of polygenic scores for the reference population compared to cases and controls in the target population. Presumably the reference population should have a distribution intermediary to cases and controls or more closely aligning to controls.

Author Reply: We appreciate the reviewer's comments which helped improve the manuscript. Since the study population used to estimate effect size of polygenic scores is enriched for disease (case-control), it is not a fair representation of the general population to define a distribution of polygenic scores for generation of percentiles. It is for this case that we elected to use a separate reference population of 1,017 participants who are not enriched for cardiometabolic disease and are representative of the Saudi population including the diversity of tribal representation from around the country.

In the revised manuscript, we show the validity of our approach using the reviewer's suggestions. We show that the PC distribution of the reference population is similar to the case-control population. We then show that the reference population's distribution of polygenic scores is similar to the control population in the study.

Manuscript Changes: We added two supplementary figures to demonstrate the validity of the reference population through PC analysis and to show a comparison of the raw distribution of polygenic scores for the reference population compared to the cases and controls in the target population.

Supplementary Figure 1. Principal component analysis comparison of Arab case-control cohort (N=5,399) to the Arab reference population (N=1,017)

Supplementary Figure 3. Raw distribution of Arab-optimized polygenic scores in reference population, cases, and controls

We added the following sentence to the Results section:

“We defined percentiles of polygenic risk based on the reference population (Supplementary Figure 1) and confirmed that the distribution of scores in the control is comparable to their distribution in the reference population (Supplementary Figure 3).”

We added the following sentence to the Online Methods section:

“This reference population had similar genetic ancestry to the case-control study population (Supplementary Figure 1), and the distribution of polygenic scores in the reference population was similar to the control (Supplementary Figure 3).”

MINOR

1. The authors reference Wikipedia (ref 13) regarding the world Arab population. A more rigorous and stable reference would strengthen the manuscript.

Author Reply: We agree and replaced the Wikipedia reference by the World Bank data.

Population, total - Arab World | Data.

<https://data.worldbank.org/indicator/SP.POP.TOTL?locations=1A>.

2. The authors state in the results, “We first asked whether the most commonly used polygenic scores for cardiometabolic disease have reduced performance in Arabs (Figure 2 a,b).” However, Figure 2a and 2b shows a map and PC analysis. Is this figure reference correct?

Author Reply: We agree and appreciate the reviewer’s careful read.

Manuscript Changes: In response, we changed the figure reference in the following sentence.

“We first asked whether the most commonly used polygenic scores for cardiometabolic disease have reduced performance in Arabs (Figure 2).”

3. Supp Table 9 shows cohort characteristics that are related to or redundant to data shown in Table 1. It may make sense to simply have one table that is referenced throughout.

Author Reply: We agree with the reviewer that this information is redundant.

Manuscript Changes: In response, we removed Supplementary Table 9 from the revised manuscript and consolidated with Table 1.

REVIEWER COMMENTS

Reviewer #1 (Remarks to the Author):

The authors responded to the review thoroughly and adequately. Nice work!

Reviewer #2 (Remarks to the Author):

Re: Clinical utility of polygenic scores for cardiometabolic disease in Arabs

In this revised manuscript, Shim and colleagues have thoroughly worked to address my previous comments, and I believe the manuscript is improved and easier to follow. I only have two remaining concerns that are related and that I think the authors can easily address.

1. It is hard to follow what subjects are used for what analyses, and often the numbers do not add up. Table 1 indicates the study population consists of 5,399 subjects, including 3,491 CAD cases and 2,979 T2DM cases. The texts suggest that application of existing PGS to this cohort is shown in Figure 2c. But this figure indicates 1,712 CAD cases and CAD 2,682 controls (total 4,394) and 1,468 T2DM cases and T2DM 2,673 controls (total 4,141). Thus, I assume some subjects were excluded from this analysis. Also, Supplementary Table 2 shows results of applications of polygenic scores to an Arab cohort. The sample size for CAD is listed as 2,682 and for T2DM as 2,673. It looks to me that this table is actually showing the results shown in Figure 2c, but the sample sizes ("N Total") are incorrect. Finally, Table 2 suggests that sample size for the validation cohort analyses is 2,699. But the first line of Figure 3 lists the total sample size for CAD analysis as 2,682. The authors should make it clear/explicit what sample is being used for each analysis, and where appropriate what are the total number of cases and controls. If subjects are being excluded, this should be explicit. A detailed study cohort diagram may be helpful.

2. Part of my confusion described above is also attributed to the reported effect sizes of the polygenic scores. For example, for CAD, the authors report an OR (95% CI) of 1.41 (1.31-1.50) when applied to the study cohort shown in Figure 2c (4,394 in total sample size). This same results is reported in Supplementary Table 2 for the study cohort that is N Total 2,682. This same result is also reported for the validation sample in Table 2 (N total 2,699 according to the legend). In fact, I think all of the Arab results in Table 2 and Supplementary Table 2 are identical, but based on the main text, I believe Table 2 should be showing the validation cohort and Supp. Table 2 should be showing the full cohort. The authors should go over each table and figure carefully and make sure the appropriate results are being reported for the appropriate analytic cohort, and as requested above, it should be clear what analytic cohort is being used along with the sample size.

Response to Reviewers' Comments for NCOMMS-23-01983A
"Clinical utility of polygenic scores for cardiometabolic disease in Arabs"

Reviewer #1 (Remarks to the Author):

The authors responded to the review thoroughly and adequately. Nice work!

Author Reply: We appreciate the reviewer's helpful comments.

Reviewer #2 (Remarks to the Author):

Re: Clinical utility of polygenic scores for cardiometabolic disease in Arabs

In this revised manuscript, Shim and colleagues have thoroughly worked to address my previous comments, and I believe the manuscript is improved and easier to follow. I only have two remaining concerns that are related and that I think the authors can easily address.

Author Reply: We are grateful for the reviewer's thoughtful review which helped improve our manuscript.

1. It is hard to follow what subjects are used for what analyses, and often the numbers do not add up. Table 1 indicates the study population consists of 5,399 subjects, including 3,491 CAD cases and 2,979 T2DM cases. The texts suggest that application of existing PGS to this cohort is shown in Figure 2c. But this figure indicates 1,712 CAD cases and CAD 2,682 controls (total 4,394) and 1,468 T2DM cases and T2DM 2,673 controls (total 4,141). Thus, I assume some subjects were excluded from this analysis. Also, Supplementary Table 2 shows results of applications of polygenic scores to an Arab cohort. The sample size for CAD is listed as 2,682 and for T2DM as 2,673. It looks to me that this table is actually showing the results shown in Figure 2c, but the sample sizes ("N Total") are incorrect. Finally, Table 2 suggests that sample size for the validation cohort analyses is 2,699. But the first line of Figure 3 lists the total sample size for CAD analysis as 2,682. The authors should make it clear/explicit what sample is being used for each analysis, and where appropriate what are the total number of cases and controls. If subjects are being excluded, this should be explicit. A detailed study cohort diagram may be helpful.

Author Reply: We appreciate how the presentation of the sample size in different parts of the manuscript might have seemed confusing, but confirm that there are no errors in the actual data or analysis in this version of the manuscript. To clarify, the total sample size of the cohort is 5,399 which was equally split into a training set (N=2,700) and a validation set (N=2,699). Upon the recommendation of the reviewers' initial revision, we presented the main analysis for optimized scores (e.g. Fig 2c, Table 2) for the validation set (N=2,699) rather than for the entire cohort (N=5,399). For each disease or trait analyzed, there was a degree of missingness in the phenotype, and as such the N total of the specific

trait in the validation set is slightly lower than 2,699. For example, there are 17 individuals in the validation set that are missing CAD phenotype ascertainment, and as such the total N (for the CAD phenotype analysis) is 2,682. We appreciate how this could be confusing unless stated clearly. Specifically in Figure 2c, the reviewer misread “N total” as “N control” which also led to additional confusion.

Manuscript Changes: In response, we have clarified the total sample size per disease/trait in the validation set in three ways:

First, we added the following sentence to the legend of Figure 2c:

“N total is the total number of samples in the validation dataset excluding missing values for each disease.”

Second, we specified the exact N total for each disease or trait after excluding missingness in Supplementary Table 5 as shown below:

Supplementary Table 5. Characteristics of training and validation datasets

Characteristic	Training dataset (N=2,700)	N total after excluding missing values	Validation dataset (N=2,699)	N total after excluding missing values
Sex, male (%)	1733 (64.2)	2,700	1740 (64.5)	2,699
Age, mean (SD)	54.95 (14.80)	2,700	54.72 (14.84)	2,699
Coronary artery disease, n (%)	1779 (66.2)	2,688	1712 (63.8)	2,682
Type 2 diabetes, n (%)	1511 (56.2)	2,687	1468 (54.9)	2,673
Cardiomyopathy, n (%)	244 (9.5)	2,581	235 (9.1)	2,590
LDL cholesterol, mean (SD) [mg/dL]	136.66 (54.13)	2,246	134.97 (52.61)	2,201
HDL cholesterol, mean (SD) [mg/dL]	45.32 (13.02)	2,251	45.08 (13.46)	2,208
Triglycerides, mean (SD) [mg/dL]	166.39 (110.03)	2,260	165.05 (113.75)	2,212
Systolic blood pressure, mean (SD) [mmHg]	147.23 (23.81)	2,606	145.99 (24.02)	2,613
Diastolic blood pressure, mean (SD) [mmHg]	83.21 (13.05)	2,606	82.90 (13.62)	2,613
Body mass index, mean (SD) [kg/m ²]	29.39 (6.08)	2,570	29.23 (6.02)	2,553
Height, mean (SD) [m]	1.61 (0.09)	2,568	1.61 (0.09)	2,553

Third, we added the following description to the legend of Table 2:

“The number of missing values varies across the disease or trait. Additional details are shown in Supplementary Table 5.”

2. Part of my confusion described above is also attributed to the reported effect sizes of the polygenic scores. For example, for CAD, the authors report an OR (95% CI) of 1.41 (1.31-1.50) when applied to the study cohort shown in Figure 2c (4,394 in total sample size). This same result is reported in Supplementary Table 2 for the study cohort that is N Total 2,682. This same result is also reported for the validation sample in Table 2 (N total 2,699 according to the legend). In fact, I think all of the Arab results in Table 2 and Supplementary Table 2 are identical, but based on the main text, I believe Table 2 should be showing the validation cohort and Supp. Table 2 should be showing the full cohort. The authors should go over each table and figure carefully and make sure the appropriate results are being reported for the appropriate analytic cohort, and as requested above, it should be clear what analytic cohort is being used along with the sample size.

Author Reply: We confirm that the reviewer is correct – the results are identical and consistent. As we stated above, part of the confusion is from misreading “N total” as “N control” in Figure 2c. In other words, the total is indeed 2,682 in Figure 2c and not 4,394. In the revised manuscript, we now clarify adding a line to the legend to Figure 2 as well as additional columns to Supplementary Table 2 describing the missingness, as detailed above. Also as we stated above, we have presented the score performance results in the validation set only following the recommendations from the initial revision. We are concerned that presenting results for the validation set, and then the similar results (i.e. a different version of the current Table 2) for the entire cohort would result in even more confusion. We agree with the reviewer's point that the dataset used for the Supplementary Table 2 results should be presented more clearly in the main text.

Manuscript Changes: We have modified the following sentence in the Results section:

“For each of the 10 traits of interest, we used one-to-one nearest neighbor matching to generate an ~~equal-sized~~ cohort of individuals of European ancestry from the UK Biobank **of equal size as the validation dataset of Arabs**, with comparable age, sex, and case-control ratio for categorical matching traits or mean value for continuous matching traits (Online Methods, Supplementary Table 2).”